RESEARCH COMMUNICATION

# *Plasmodium* Niemann-Pick type C1-related protein is a druggable target required for parasite membrane homeostasis

Eva S Istvan[1,2†], Sudipta Das[3†], Suyash Bhatnagar[3], Josh R Beck[1,2‡], Edward Owen[4,5,6], Manuel Llinas[4,5,6], Suresh M Ganesan[7], Jacquin C Niles[7], Elizabeth Winzeler[8], Akhil B Vaidya[3*], Daniel E Goldberg[1,2*]

[1]Department of Medicine, Division of Infectious Diseases, Washington University School of Medicine, Saint Louis, United States; [2]Department of Molecular Microbiology, Washington University School of Medicine, Saint Louis, United States; [3]Department of Microbiology and Immunology, Center for Molecular Parasitology, Drexel University College of Medicine, Philadelphia, United States; [4]Department of Biochemistry and Molecular Biology, Pennsylvania State University, University Park, United States; [5]Huck Center for Malaria Research, Pennsylvania State University, University Park, United States; [6]Department of Chemistry, Pennsylvania State University, University Park, United States; [7]Department of Biological Engineering, Massachusetts Institute of Technology, Cambridge, United States; [8]Department of Pediatrics, University of California San Diego School of Medicine, La Jolla, United States

*For correspondence:
av27@drexel.edu (ABV);
dgoldberg@wustl.edu (DEG)

†These authors contributed equally to this work

Present address: ‡Department of Biomedical Science, Iowa State University, Ames, United States

Competing interests: The authors declare that no competing interests exist.

**Abstract** *Plasmodium* parasites possess a protein with homology to Niemann-Pick Type C1 proteins (Niemann-Pick Type C1-Related protein, NCR1). We isolated parasites with resistance-conferring mutations in *Plasmodium falciparum* NCR1 (PfNCR1) during selections with three diverse small-molecule antimalarial compounds and show that the mutations are causative for compound resistance. PfNCR1 protein knockdown results in severely attenuated growth and confers hypersensitivity to the compounds. Compound treatment or protein knockdown leads to increased sensitivity of the parasite plasma membrane (PPM) to the amphipathic glycoside saponin and engenders digestive vacuoles (DVs) that are small and malformed. Immuno-electron microscopy and split-GFP experiments localize PfNCR1 to the PPM. Our experiments show that PfNCR1 activity is critically important for the composition of the PPM and is required for DV biogenesis, suggesting PfNCR1 as a novel antimalarial drug target.
**Editorial note:** This article has been through an editorial process in which the authors decide how to respond to the issues raised during peer review. The Reviewing Editor's assessment is that all the issues have been addressed (see decision letter).
DOI: https://doi.org/10.7554/eLife.40529.001

## Introduction

Several whole-parasite chemical library screens have identified thousands of compounds with potent antimalarial activity (*Guiguemde et al., 2010*; *Kato et al., 2016*). To facilitate drug development, it is important to identify targets of these compounds. Target identification can be extremely challenging, especially in organisms like *Plasmodium* that contain large numbers of proteins with unknown

function. Evolution of compound-resistant malaria parasites can be helpful in the discovery of the molecular mechanisms by which compounds kill the organism (*Rathod et al., 1994*; *Rottmann et al., 2010*; *Vaidya et al., 2014*; *Istvan et al., 2017*).

In this study, we investigated a gene that acquired single nucleotide polymorphisms (SNPs) or was amplified in selections with three diverse compounds. PF3D7_0107500 encodes a membrane protein with sequence motifs found in Niemann-Pick C1 (NPC1) proteins. Human NPC1 (hNPC1) has been the subject of numerous studies because of the protein's importance in cholesterol egress from late endosomes (*Pentchev, 2004*). Patients with mutations in hNPC1 suffer a fatal neurodegenerative lipid storage disorder characterized by the accumulation of lysosomal cholesterol, sphingomyelin, as well as other lipids (*Gong et al., 2016*). Niemann-Pick C1-Related (NCR1) proteins are conserved in eukaryotic evolution and are most easily identified by their membrane domains (*Higaki et al., 2004*). In humans, NPC1 accepts cholesterol from its partner protein, the high affinity cholesterol-binding protein NPC2 (*Li et al., 2016*). NCR1 homologs are also present in organisms that do not contain readily identifiable NPC2 proteins or internalize sterol by endocytosis. Based on studies with yeast NCR1, Munkacsi et al. proposed that the primordial function of NCR1 is the regulated transport of lipophilic substrates such as sphingolipids (*Munkacsi et al., 2007*).

Until now the function of PF3D7_0107500, which we call *Plasmodium falciparum* Niemann-Pick Type C1-Related protein (PfNCR1), has been unclear. In this study, we prepared a genetic knockdown (K/D) of *pfncr1* and showed that K/D critically slows blood-stage parasite replication. Furthermore, *pfncr1* K/D caused parasites to become abnormally sensitive to the pore-forming amphipathic glycoside saponin. Treatment with any of the three compounds that we identified during resistance selection phenocopied the gene K/D, suggesting that the compounds interfere with PfNCR1 function. Here we show that PfNCR1 is druggable and necessary for maintaining the proper membrane lipid composition of blood-stage parasites.

## Results

### Mutations in PfNCR1 provide resistance to three diverse compounds

As part of a study aimed at analyzing the *P. falciparum* resistome (*Corey et al., 2016*), we isolated parasites resistant to three structurally diverse compounds with similar, submicromolar potencies against wild-type parasites (*Figure 1A* and *Figure 1—source data 1*). Resistant parasites contained mutations in one common gene, PF3D7_0107500, which is predicted to encode a 1470 amino acid membrane protein. Sequence similarity searches indicated homology to a protein previously studied in the related apicomplexan parasite *Toxoplasma gondii* called Niemann-Pick Type C1-Related Protein (TgNCR1). Lige et al. identified sequence elements conserved between TgNCR1 and hNPC1, a lysosomal integral membrane protein (*Lige et al., 2011*). The same sequence elements are also present in PfNCR1. Cryo-EM and crystal structures of hNPC1 reveal a 13-helix transmembrane region containing a sterol-sensing domain (SSD) (orange) and a conserved C-terminal transmembrane domain (C-TM) (magenta) (*Figure 1B*) (*Gong et al., 2016*; *Li et al., 2016*). The C-terminal targeting sequence that extends past the C-TM in hNPC1 and localizes this protein to the lysosome, is not present in PfNCR1. Lumen-exposed domains (grey and blue in *Figure 1B*) complete the hNPC1 structure. Sequence similarity between hNPC1 and PfNCR1 is restricted to portions of the transmembrane region (orange and magenta) and to approximately 45 amino acids N-terminal to the SSD (red). Based on this limited sequence similarity, we generated a cartoon model of PfNCR1 (*Figure 1C*). We observed five mutations in our compound-resistant parasites: A1108T came from selections with MMV009108; M398I and A1208E from selections with MMV028038; and S490L and F1436I from selections with MMV019662. The model suggests that three of the mutations are proximal to the membrane domain, while the other two localize to the hydrophilic domains. We used single-crossover allelic exchange to introduce one mutation from each resistance selection into a clean genetic background (*Figure 1—figure supplements 1–2*). With this strategy, PfNCR1 is expressed from its native promoter and contains a C-terminal green-fluorescent protein (GFP) tag in addition to the mutation. We also generated non-mutated allelic exchange control parasites containing the GFP tag. Inclusion of the C-terminal GFP did not alter the sensitivity to MMV009108 (*Figure 1D*), while parasites with single mutations in PfNCR1 were resistant to the compounds with which they were selected (*Figure 1D–F*, *Figure 1—source data 2*).

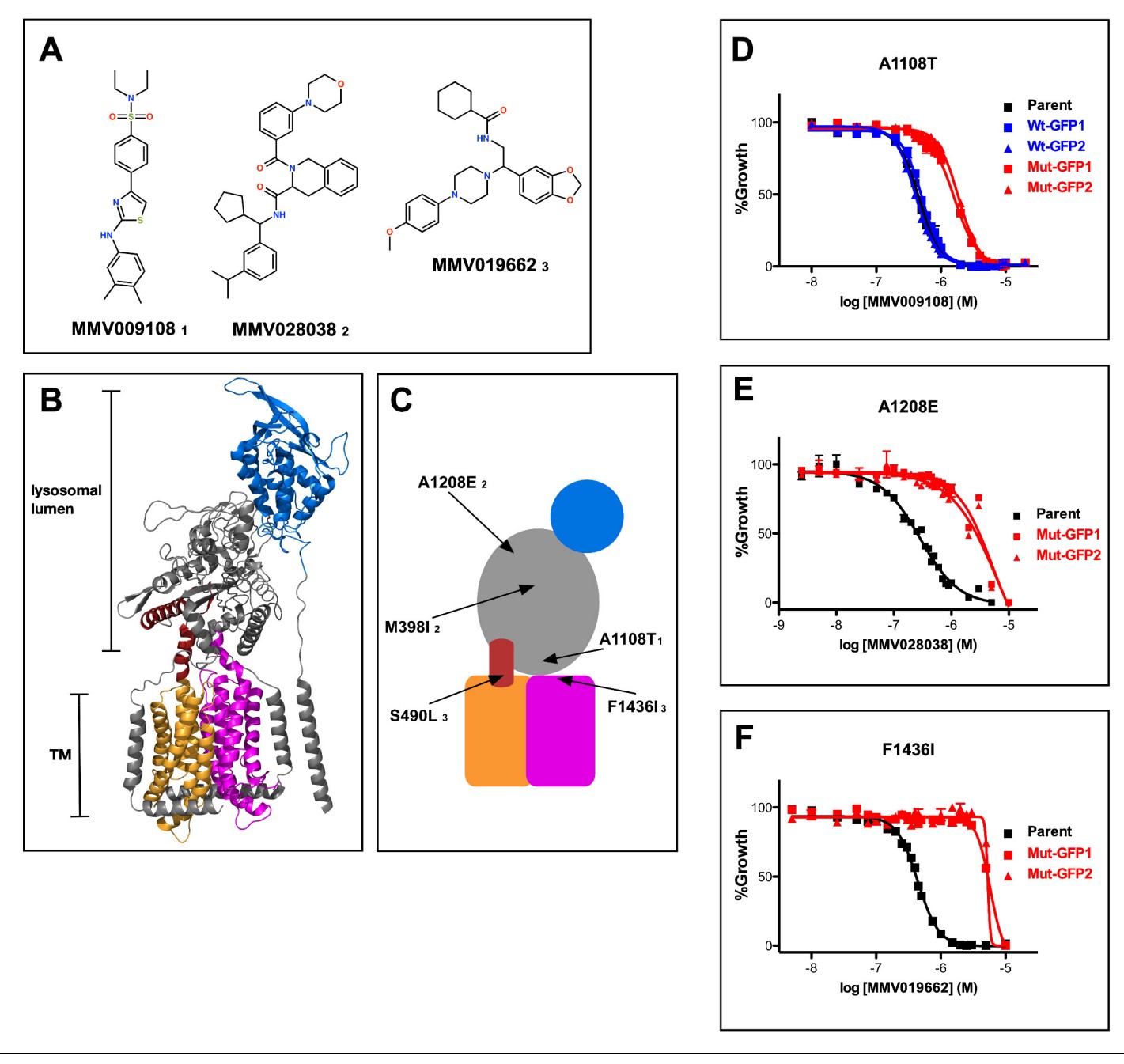

**Figure 1.** Mutations in PfNCR1 confer resistance to three antimalarial compounds. (A) Structures of the three structurally diverse compounds that yielded mutations in PfNCR1. The lower-case numbers next the compound IDs are used in C) to match mutations with specific compounds. (B) Ribbon model of the structure of hNCR1 solved by cryoEM (*Gong et al., 2016*). PDB coordinates: 3JD8. The SSD is shown in orange, the conserved C-terminal membrane domain is shown in magenta, the domain that interacts with hNCR2 is in blue and an additional sequence stretch with similarity to PfNCR1 is in red. (C) Cartoon model of the possible domain arrangement in PfNCR1. Sequence similarity to hNCR1 is restricted to the red, orange and magenta domains. Locations of resistance-conferring mutations are shown with arrows. Compound IDs matching with mutations are shown in lower case numbers and match *Figure 1A*. The model was generated by visual examination of the hNCR1 structure, aided by the alignment of hNCR1 aa 580–794 and aa 1083–1253 with PfNCR1 aa 439–662 and aa 1304–1468, and aided by a partial model of C-terminal residues generated by Robetta (*Ovchinnikov et al., 2018*). (D–F) Concentration response curves of blood-stage parasites (all in 0.1% DMSO) measured using a flow cytometry-based assay. Each panel shows different compound and a different mutation. (D) MMV009108, (E) MMV028038, (F) MMV019662. Black = parental 3D7 parasites; red = two independent clones of parasites with mutant allelic exchange; blue = two independent clones of parasites with wild-type allelic exchange (for part D only). The error bars (S.D.) for a representative experiment (technical triplicates) are shown and are very small. The experiment in D) was done three times, (E) and (F) were done twice. For each one representative experiment is shown.

*Figure 1 continued on next page*

*Figure 1 continued*

DOI: https://doi.org/10.7554/eLife.40529.002

The following source data and figure supplements are available for figure 1:

**Source data 1.** Potencies of compounds against parental (wild-type) parasites in *Figure 1D–F*.
DOI: https://doi.org/10.7554/eLife.40529.006
**Source data 2.** Resistance of allelic exchange-modified parasites, compared to parental parasites.
DOI: https://doi.org/10.7554/eLife.40529.007
**Figure supplement 1.** Characterization of 1108 allelic exchange clones.
DOI: https://doi.org/10.7554/eLife.40529.003
**Figure supplement 2.** Characterization of A1208E and F1436I clones.
DOI: https://doi.org/10.7554/eLife.40529.004
**Figure supplement 3.** Concentration response curves of PfNCR1 mutant parasite clones with compounds.
DOI: https://doi.org/10.7554/eLife.40529.005

We examined the effect of single mutations on the different compounds. A1108T or F1436I mutant parasites were resistant to all three compounds, while A1208E mutant parasites were sensitive to the two compounds that were not used in the A1208E selection (*Figure 1D–F*, *Figure 1—figure supplement 3*, *Figure 1—source data 2*). These findings suggest that amino acids modeled to be proximal to the membrane domain (A1108 and F1436) may have some functional overlap, while the putative soluble domain A1208 may have a different activity.

## PfNCR1 is important for asexual parasite viability and is targeted by antimalarial compounds

An attempt to disrupt the *pfncr1* gene using a CRISPR/Cas9-targeting approach did not succeed, suggesting an essential function during blood-stage malaria growth. Next, we created parasites in which *pfncr1* expression is regulated by anhydrotetracyline (aTc) using the previously described TetR-DOZI/aptamer translational repression technology (*Ganesan et al., 2016*; *Spillman et al., 2017*) (*Figure 2—figure supplement 1A–C*). When we removed aTc from highly synchronized, young ring-stage parasites, PfNCR1 expression in trophozoites was reduced within the same cell cycle and undetectable in the following cell cycles, as judged by western blots to detect a C-terminal hemagglutinin (HA) sequence on the aptamer-tagged parasites (*Figure 2A*). While protein levels after aTc withdrawal were affected almost immediately, parasite replication rates decreased only after 3–4 days (*Figure 2B*, inset). After this slow onset of reduced growth, PfNCR1 K/D clearly resulted in markedly less fit parasites. Essentiality of NCR1 in *Plasmodium* is further supported by a mutagenesis study in *P. falciparum* (*Zhang et al., 2018*) and by a *P. berghei* knockout study (*Bushell et al., 2017*). Complementing K/D parasites with a second copy wild-type PfNCR1 rescued the growth defect (*Figure 2C*, *Figure 2—figure supplement 1D*). Modulating the expression level of PfNCR1 with aTc shifted the MMV009108 concentration-response curve (*Figure 2D*) and maximal K/D hypersensitized parasites to the three compounds that were used for the resistance selection (*Figure 2E–H*, *Figure 2—source data 1*). Our findings suggest that PfNCR1 performs a function important for the viability of blood-stage malaria parasites and that the three compounds act directly on the protein.

## PfNCR1 localizes to the parasite plasma membrane

To better understand the functional significance of PfNCR1, we localized the protein. For this purpose, we used parasites expressing wild-type PfNCR1 protein tagged with a C-terminal GFP from its native promoter. Live microscopy showed fluorescence surrounding the intraerythrocytic parasites (*Figure 3A*). The distribution of GFP was in contrast to an earlier suggestion that PfNCR1 may reside in the digestive vacuole (DV) membrane (*Martin et al., 2009*). Immuno-electron microscopy of parasites expressing GFP or HA-tagged PfNCR1 confirmed localization of the protein to the membranes surrounding parasites (*Figure 3B,C*, *Figure 3—figure supplement 1*). Blood-stage parasites are surrounded by two membranes in very close apposition - the parasitophorous vacuolar membrane (PVM) and the PPM. The resolution of our immuno-electron microscopy images was not sufficient to definitively determine whether PfNCR1 is present in the PVM or the PPM. To answer this question,

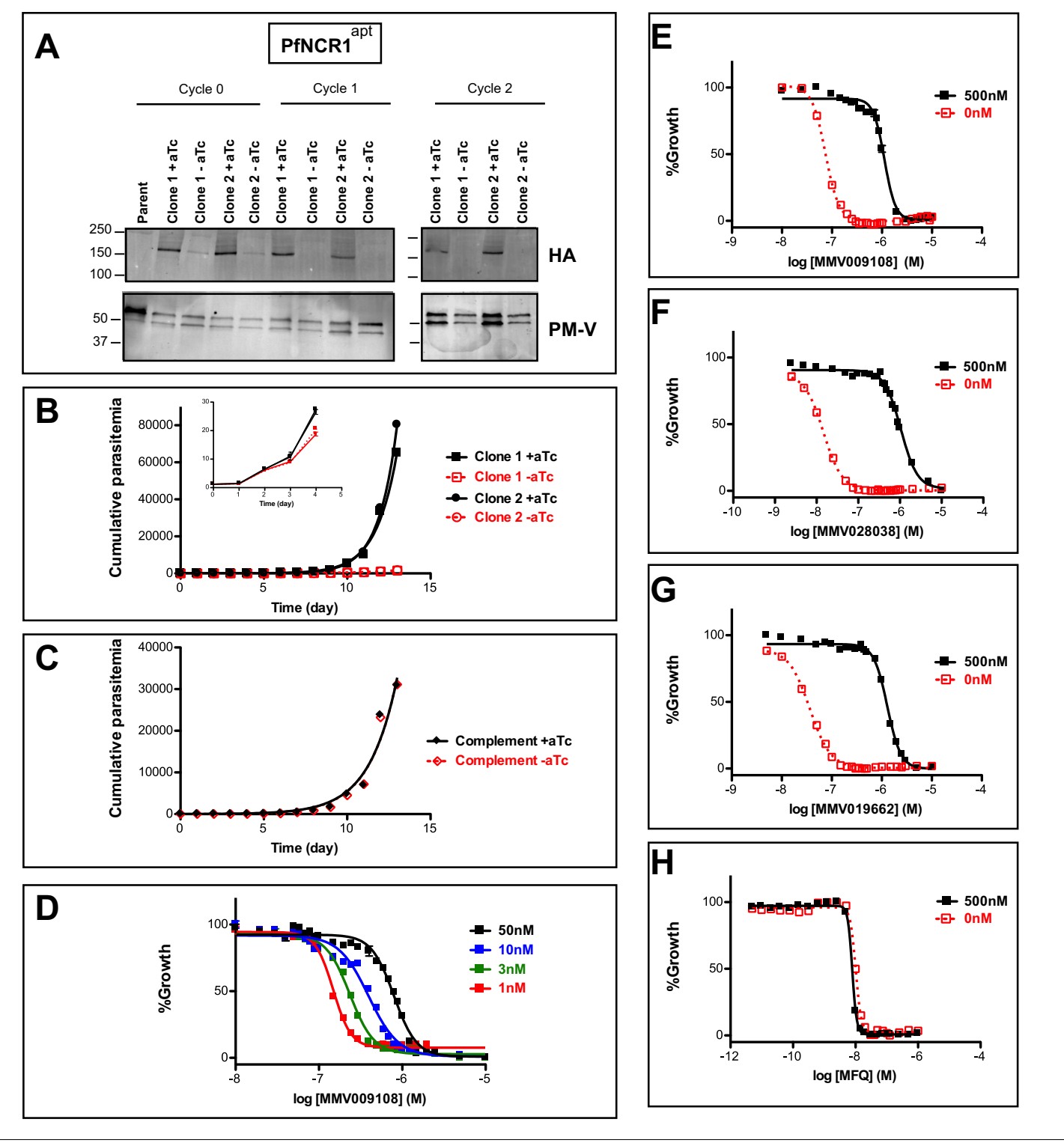

**Figure 2.** PfNCR1 is required for blood-stage parasite replication and is targeted by three antimalarials. (**A**) Western blot showing regulation of the PfNCR1apt by aTc. Trophozoite-stage parasites were harvested from the replication cycle in which aTc was removed (cycle 0), as well as the following two cycles. PfNCR1 was detected using a C-terminal HA-tag. The ER membrane protein plasmepsin V (PM-V) was used as a loading control. Note that the two bands recognized by α-PM-V antibody correspond to the full-length protein and a proteolytic fragment of the protein produced during the membrane isolation. Expected sizes: 171 k Da for PfNCR1-HA, 69 k Da for PM-V. This experiment was done two times. (**B**) Replication of PfNCR1apt parasites. Using a flow cytometry assay, the replication of two PfNCR1apt clones was monitored over two weeks. +aTc is in black and solid lines, -aTc is

*Figure 2 continued on next page*

*Figure 2 continued*

in red and dashed lines. Cultures were seeded at 1% parasitemia, and subjected to daily media changes, and/or sub-culturing. Cumulative parasitemias were calculated by multiplying with dilution factors. One representative experiment with technical triplicates is shown. The inset magnifies the initial time points. Doubling times in days are as follows (95% confidence intervals in parentheses): clone 1 -aTc = 1.596 (1.546–1.650), $R^2$ = 0.9964; clone 1 +aTc = 0.8663 (0.8218–0.9159), $R^2$ = 0.9930; clone 2 -aTc = 1.463 (1.417–1.512), $R^2$ = 0.9965; clone 2 +aTc = 0.7776 (0.7559–0.8005), $R^2$ = 0.9981. This experiment was done four times. (C) Complementation of PfNCR1$^{apt}$ rescues growth phenotype. Wild-type PfNCR1 was stably expressed in the PfNCR1$^{apt}$ background. Replication of parasites was monitored over two weeks. +aTc is in black and solid line, -aTc is in red and dashed line. One representative experiment with technical triplicates is shown. Doubling times in days are as follows (95% confidence intervals in parentheses): -aTc = 1.152 (1.036–1.298), $R^2$ = 0.97; +aTc = 1.166 (1.039–1.329), $R^2$ = 0.96. This experiment was done four times. Note that the complemented strain grows less well than PfNCR1$^{apt}$ with aTc (B), but that there is no significant difference ±aTc. (D) Expression level of PfNCR1 correlates with sensitivity to MMV009108. Concentration response curves using a flow cytometry-based growth assay. After aTc washout, aTc was replenished in triplicate cultures at different concentrations and parasitemias were measured after 72 hr. aTc concentrations are indicated. This experiment was done three times. (E–H) PfNCR1 K/D hypersensitizes parasites to three compounds. Concentration-responses of PfNCR1$^{apt}$ parasites to E) MMV009108, (F) MMV028038, (G) MMV019662, and H) mefloquine (MFQ) (control compound) without aTc (red open symbols, dashed lines) or with 500 nM aTc (black symbols, solid lines) after 72 hr. One representative experiment with technical triplicates is shown. The experiment in E) was done three times the experiments in F-H) were done two times.

DOI: https://doi.org/10.7554/eLife.40529.008

The following source data and figure supplement are available for figure 2:

**Source data 1.** Shifts in EC$_{50}$s under PfNCR1 K/D.
DOI: https://doi.org/10.7554/eLife.40529.010
**Figure supplement 1.** Editing of the PfNCR1 locus to generate aptamer-regulated strains and generation of the PfNCR1 complementation.
DOI: https://doi.org/10.7554/eLife.40529.009

we prepared split-GFP constructs (*Cabantous et al., 2005*; *Külzer et al., 2013*) in which GFP strands 1–10 are expressed either in the parasite cytoplasm or targeted to the lumen of the parasitophorous vacuole (PV) and GFP strand 11 is expressed as a C-terminal tag on PfNCR1 (*Figure 3—figure supplement 2*). GFP fluorescence was only observed when cytoplasmic GFP 1–10 was co-expressed with PfNCR1-GFP11 (*Figure 3D–G*), suggesting that the C-terminal residues of PfNCR1 project into the parasite cytoplasm. We cannot rule out the possibility that the cytoplasmic GFP 1–10 signal is due to vesicles at the PPM in transit to the PVM, but based on these results, we propose a model in which PfNCR1 membrane domains are in the PPM while the soluble domains project into the PV (*Figure 3H*). In contrast to hNPC1, PfNCR1 does not appear to localize to internal organellar membranes. Nevertheless, our model suggests that the relative orientation of the cytosolic regions in these two distantly related proteins is conserved.

## Compound treatment or protein knockdown hypersensitizes parasites to saponin

Saponins are amphipathic glycosides with high affinity for cholesterol that are capable of penetrating membranes (*Seeman et al., 1973*; *Gögelein and Hüby, 1984*). Inhibiting the Na$^+$-efflux pump PfATP4 has previously been shown to lead to changes in PPM saponin sensitivity (*Das et al., 2016*). We were curious whether interfering with PfNCR1 function would have similar effects. We noticed decreased levels of cytosolic aldolase protein in saponin parasite extracts after incubation with MMV009108, while levels of the PVM-localized membrane-bound protein EXP2 did not change (*Figure 4A*). Hypersensitivity to saponin was reversed when MMV009108 was removed by washout. We obtained similar results in experiments probing for a different cytosolic protein, haloacid dehalogenase 1 (HAD1) (*Figure 4—figure supplement 1A*). Using a flow cytometry-based assay and a previously reported parasite clone expressing eGFP (*Straimer et al., 2012*), we observed elevated saponin-induced leakage of cytoplasmic eGFP after incubation of parasites with sub-EC50 concentrations of MMV009108, MMV028038 and MMV019662 (*Figure 4B–D*). Western blots probing for eGFP in supernatant and pellet fractions showed that the decrease in signal of cytosolic proteins was not a consequence of increased protein degradation, but rather of elevated leakage of cytoplasmic contents (*Figure 4—figure supplement 1B*). These results suggest that the PPM, the membrane to which PfNCR1 localizes, undergoes a redistribution of membrane lipids during compound treatment. We have previously shown that, for PfATP4 inhibitors, induction of saponin sensitivity is abrogated in parasites adapted to grow in low [Na$^+$] (*Das et al., 2016*). This was different from the effect of MMV009108 treatment where we observed saponin hypersensitivity in regular medium, as

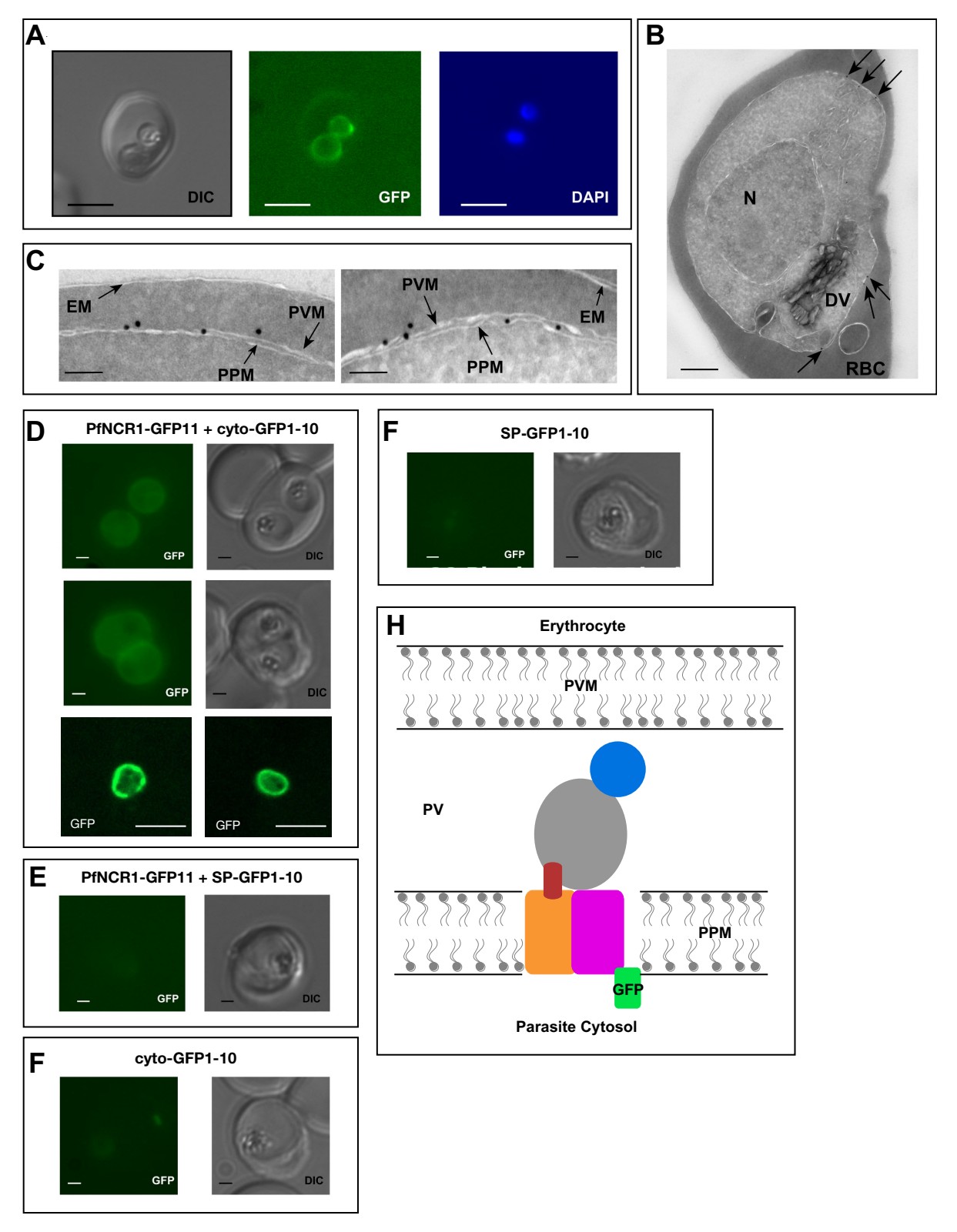

**Figure 3.** PfNCR1 localizes to the parasite plasma membrane. (**A**) Live fluorescence microscopy with C-terminally GFP-tagged wild-type PfNCR1-expressing parasites (clone Wt-GFP1 from *Figure 1*) localizes PfNCR1 to the parasite surface. Scale bar 5 μm. (**B–C**) Immuno-electron-micrographs of trophozoite-stage parasites using α-GFP antibody. Arrows mark gold particles, RBC = infected red blood cell, DV = digestive vacuole, N = nucleus. The close-up in C) shows gold particles clustered at the parasite-delimiting membranes. EM = erythrocyte membrane; PVM = parasitophorous vacuolar

*Figure 3 continued on next page*

Figure 3 continued

membrane; PPM = parasite plasma membrane. Scale bar B = 500 nm, C = 100 nm. (**D–G**) Live fluorescence microscopy on split-GFP expressing parasites. (**D**) Co-expression of PfNCR1-GFP11 with cytoplasmic GFP1-10. The bottom panels were generated using confocal microscopy. (**E**) Co-expression of PfNCR1-GFP11 with GFP1-10 that contains a signal peptide and localizes to the vacuole. (**F**) Cytoplasmic GFP1-10 without expression of PfNCR1-GFP11. (**F**) GFP-1–10 containing a signal peptide without expression of PfNCR1-GFP11. Scale bar: 1 µm for epifluorescence images, 10 µm for confocal images. (**H**) Cartoon of the proposed orientation of PfNCR1 in the PPM (parasite plasma membrane). PV = parasitophorous vacuole; PVM = parasitophorous vacuolar membrane.

DOI: https://doi.org/10.7554/eLife.40529.011

The following figure supplements are available for figure 3:

**Figure supplement 1.** Immuno-electron microscopy of HA-tagged PfNCR1.

DOI: https://doi.org/10.7554/eLife.40529.012

**Figure supplement 2.** Preparation of PfNCR1-GFP11.

DOI: https://doi.org/10.7554/eLife.40529.013

well as in low [Na$^+$]-containing medium (**Figure 4E**). Also, unlike PfATP4 inhibitors, MMV009108 did not result in Na$^+$ influx into parasites (**Figure 4F**). PfNCR1 K/D did not change sensitivity to KAE609 (**Figure 4—figure supplement 1C** and **Figure 4—source data 1**). We conclude that MMV009108 acts directly on PfNCR1 but suggest that PfATP4 activity influences PfNCR1 function (PfATP4 mutants are hypersensitive to our compounds (**Corey et al., 2016**)).

We looked for changes in the PVM using a parasite clone in which the fluorescent protein mRuby3 is targeted via a signal peptide to the PV (**Figure 5A**). As expected, the PVM was exquisitely sensitive to saponin and mRuby was released irrespective of drug treatment (**Figure 5B**). Leakage of cytosolic HAD1 after saponin treatment was enhanced by MMV009108, as previously seen (**Figure 5—figure supplement 1**). With the same PV-targeted mRuby parasites we examined the sensitivity of the PVM to the cholesterol-binding toxin tetanolysin, which, at low concentrations, normally lyses the erythrocyte membrane but not the PVM (**Hiller et al., 2003**). Treatment with MMV009108 did not alter PVM susceptibility to tetanolysin (**Figure 5C**), suggesting that compound treatment does not perturb PVM lipid composition.

Next, we examined whether PfNCR1$^{apt}$ parasites are hypersensitive to saponin after K/D. Removal of aTc sensitized parasites to saponin as monitored by the loss of cytoplasmic HAD1, while complemented control parasites expressing wild-type PfNCR1 in the K/D parasite background had normal saponin sensitivity (**Figure 6A**). As an independent marker, we prepared a PfNCR1$^{apt}$ parasite line expressing cytosolic eGFP. In this background, PfNCR1 K/D increased PPM sensitivity to saponin within 22 hr of aTc removal (**Figure 6B**), much more rapidly than the onset of slowed parasite growth (**Figure 2B**). Adding back aTc to PfNCR1$^{apt}$ parasites rapidly restored normal saponin sensitivity (**Figure 6B**). Similarly, saponin sensitivity after K/D of PfNCR1 for 40 hr was reversible in as little as 2 hr (**Figure 6—figure supplement 1**). In summary, PfNCR1 K/D phenocopies the effect of the three compounds on the PPM, suggesting that the compounds we identified interfere with PfNCR1 activity and that PfNCR1 function is required to maintain normal PPM lipid composition.

## PfNCR1 activity is required for digestive vacuole function

We hypothesized that DV formation could be affected by PfNCR1 impairment as DVs are formed from endocytic vesicles that invaginate at the PPM (**Figure 7A**). To observe DVs in live parasites we used a strain that expresses GFP as a fusion protein with the DV protease plasmepsin II (PMII) (**Klemba et al., 2004**). In this strain, PMII-GFP is produced as a membrane-bound pro-enzyme that enters the secretory pathway and is delivered from the ER to the PPM. At the PPM, pro-PMII-GFP accumulates in cytostomes and migrates via vesicles to the DV.

After incubation with compounds we noticed abnormally punctate and occasionally diffuse GFP fluorescence that was not concentrated in DVs (**Figure 7B,C**). Whereas most DMSO-treated control parasites had round DVs of ~2 µm diameter and contained only a few small submicron GFP-positive dots, compound-treated parasites frequently had many small fluorescent foci, some of which were unusually bright. To confirm that abnormal DVs were a consequence of interfering with normal PfNCR1 function, we introduced PfNCR1$^{apt}$ into the parasite line containing the PMII-GFP fusion (**Figure 7 —figure supplement 1A-C**). PfNCR1 K/D parasites had dispersed GFP puncta similar to those seen in compound-treated parasites (**Figure 7D,E**). Electron micrographs prepared from

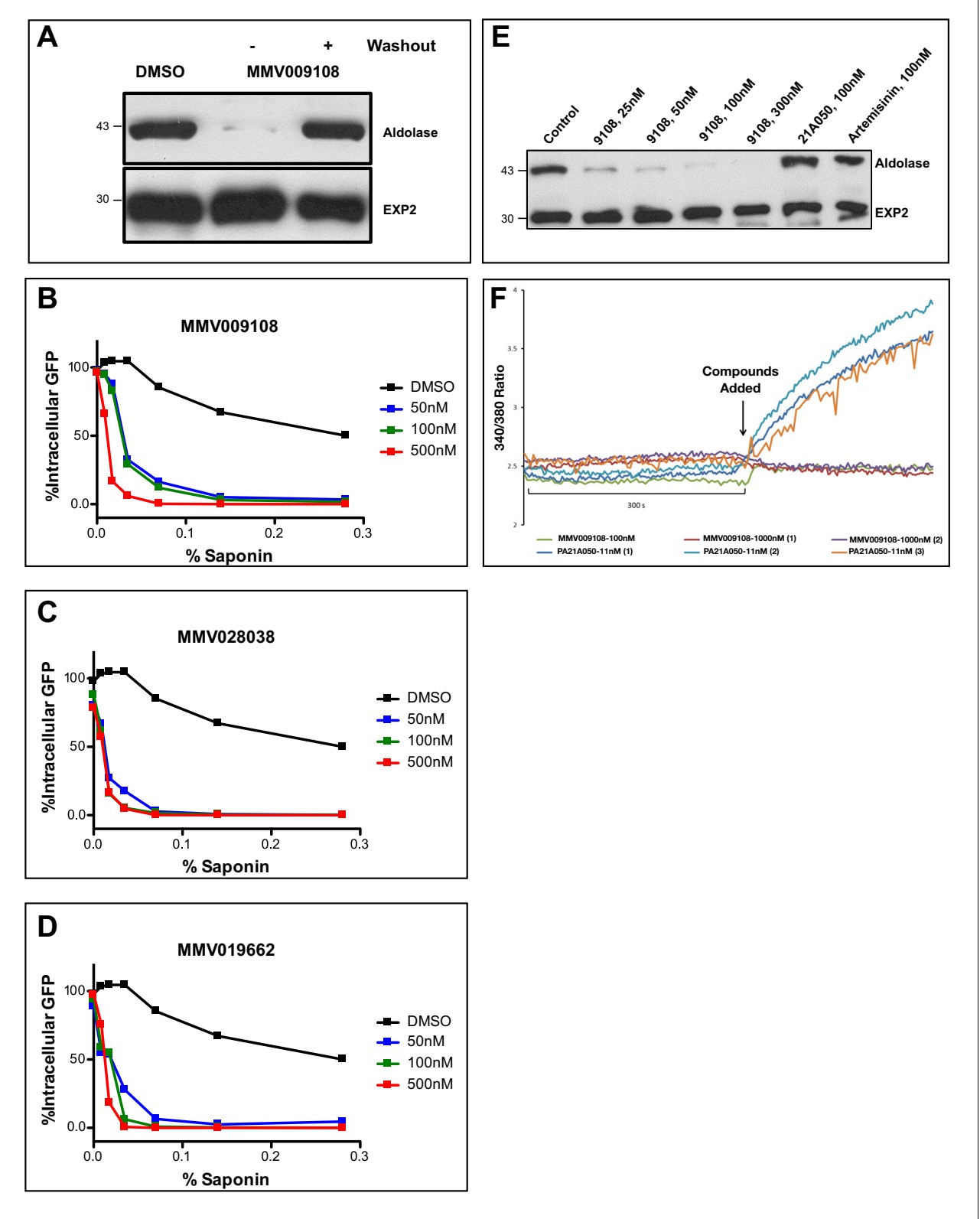

**Figure 4.** Compound treatment hypersensitizes parasites to saponin. (**A**) Strain 3D7 parasites (30–34 hr post-infection) were exposed to DMSO or 100 nM MMV009108 for 2 hr. Compound or vehicle were removed by washout and rescued by growing in compound-free cRPMI medium for another 2 hr. Parasites were treated with saponin (0.02%) to release parasites followed by western blot analysis using antibodies to parasite aldolase or EXP2. EXP2 was used as a loading control. This experiment was done three times. B – D) Flow cytometry-based assay to monitor cell leakiness using a cytoplasmic

*Figure 4 continued on next page*

*Figure 4 continued*

GFP expressing parasite clone (NF54[eGFP]). Parasites were incubated with MMV009108 (B), MMV028038 (C), or MMV019662 (D) at the indicated concentrations for 1 hr (DMSO was the vehicle control). Following compound washout with PBS, parasites were released from RBCs with saponin. Using flow cytometry, 50,000 cells were counted and scored as GFP positive or negative. At 0% saponin, all samples had similar numbers of GFP-positive cells (~80%). The experiment in B) was done three times. The experiments in C and D) were done two times. For each B)-D) a single representative experiment (with technical duplicates) is shown. E) Low Na$^+$-adapted trophozoite stage 3D7 parasites were subjected to varying concentration of MMV009108 for 2 hrs followed by saponin (0.02%) treatment to release the parasites and subjected for western blot analysis using antibodies to parasite aldolase or EXP2 (loading control). 100 nM PA21A050[24] and 100 nM artemisinin were used as controls. This experiment was done two times. Unlike the pyrazoleamide PA21A050, MMV009108 does not induce Na$^+$ influx into parasites. Low Na$^+$-adapted trophozoite stage 3D7 parasites were subjected to varying concentration of MMV009108 for 2 hrs followed by saponin (0.02%) treatment to release the parasites and subjected for western blot analysis using antibodies to parasite aldolase or EXP2 (loading control). 100 nM PA21A050[24] and 100 nM artemisinin were used as controls. This experiment was done two times.Unlike the pyrazoleamide PA21A050, MMV009108 does not induce Na$^+$ influx into parasites. SBFI 340 nm/380 nm emission ratio traces are plotted for indicated compounds and concentration. Unlike the pyrazoleamide PA21A050, MMV009108 does not alter intracellular [Na$^+$] as represented by the lack of change in SBFI 340/380 ratiometric traces. This experiment was done three times.

DOI: https://doi.org/10.7554/eLife.40529.014

The following source data and figure supplement are available for figure 4:

**Source data 1.** EC$_{50}$fold change to KAE609 under PfNCR1 K/D.

DOI: https://doi.org/10.7554/eLife.40529.016

**Figure supplement 1.** Further characterization of PfNCR1 inhibition or K/D.

DOI: https://doi.org/10.7554/eLife.40529.015

parasites under PfNCR1 K/D (*Figure 7F*) or treated with MMV009108 (*Figure 7G*) showed dramatic defects. Normal DVs are easily distinguished from the parasite cytosol, not only because they contain hemozoin crystals, but also because they are electron-lucent. In contrast, the abnormal DVs we observed were electron-dense, smaller, elongated and irregular in shape. Usually, we could see multiple hemozoin-containing vesicles in PfNCR1-depleted/inhibited parasites.

To investigate whether DV membranes might contain defects similar to those observed in the PPM after PfNCR1 K/D or compound treatment, we measured the saponin sensitivity of the DV membrane. In PMII-GFP parasites, free GFP is hydrolyzed from PMII-GFP in the DV (*Figure 7A* and ref (*Klemba et al., 2004*)). DV-resident GFP was released from drug-treated parasites at low saponin concentrations that did not affect control parasite DVs (*Figure 7H,I*). Importantly, the levels of pro-PMII-GFP did not change, suggesting that the synthesis of PMII was not affected. To control for the possibility that DV membranes have increased leakiness after compound treatment or PfNCR1 K/D simply because the PPM is leaky and less detergent is necessary to access the DV, we repeated the experiment with isolated DVs. Again, after incubation with MMV009108, low saponin concentrations resulted in leakage of DV-localized GFP (*Figure 7—figure supplement 1D*).

Metabolomic profiling of parasite extracts after incubation with MMV009108, MMV091662 (published previously (*Allman et al., 2016*)) or MMV028038 (*Figure 8*) showed reductions across hemoglobin-derived peptides, supporting the hypothesis that the normal function of the DV has been compromised by the compounds.

## Discussion

We have identified PfNCR1, Niemann-Pick C1-related protein, as a new antimalarial target that resides in the PPM and serves important functions during intraerythrocytic growth of *P. falciparum*. Through a chemical genetics approach we have provided evidence suggesting that three structurally diverse small molecules target PfNCR1. Conditional K/D of *pfncr1* gene expression resulted in parasite demise. Phenotypic consequences of compound treatment or of conditional K/D of PfNCR1 were essentially identical, strongly suggesting that the compounds directly inhibit PfNCR1.

PfNCR1 belongs to a superfamily of multi-pass transmembrane proteins involved in a variety of biological functions ranging from being receptors for signaling molecules to transport of different types of hydrophobic molecules (*Higaki et al., 2004*; *Eicher et al., 2014*; *Trinh et al., 2017*). Currently, the gene encoding this protein, PF3D7_0107500, is annotated as a lipid/sterol:H$^+$ symporter (www.plasmodb.org). However, on the basis of its sequence similarity with previously investigated proteins from *Saccharomyces cerevisiae* (ScNCR1) (*Higaki et al., 2004*) and *Toxoplasma gondii* (TgNCR1) (*Lige et al., 2011*) we believe it is more appropriate to name it as PfNCR1. When

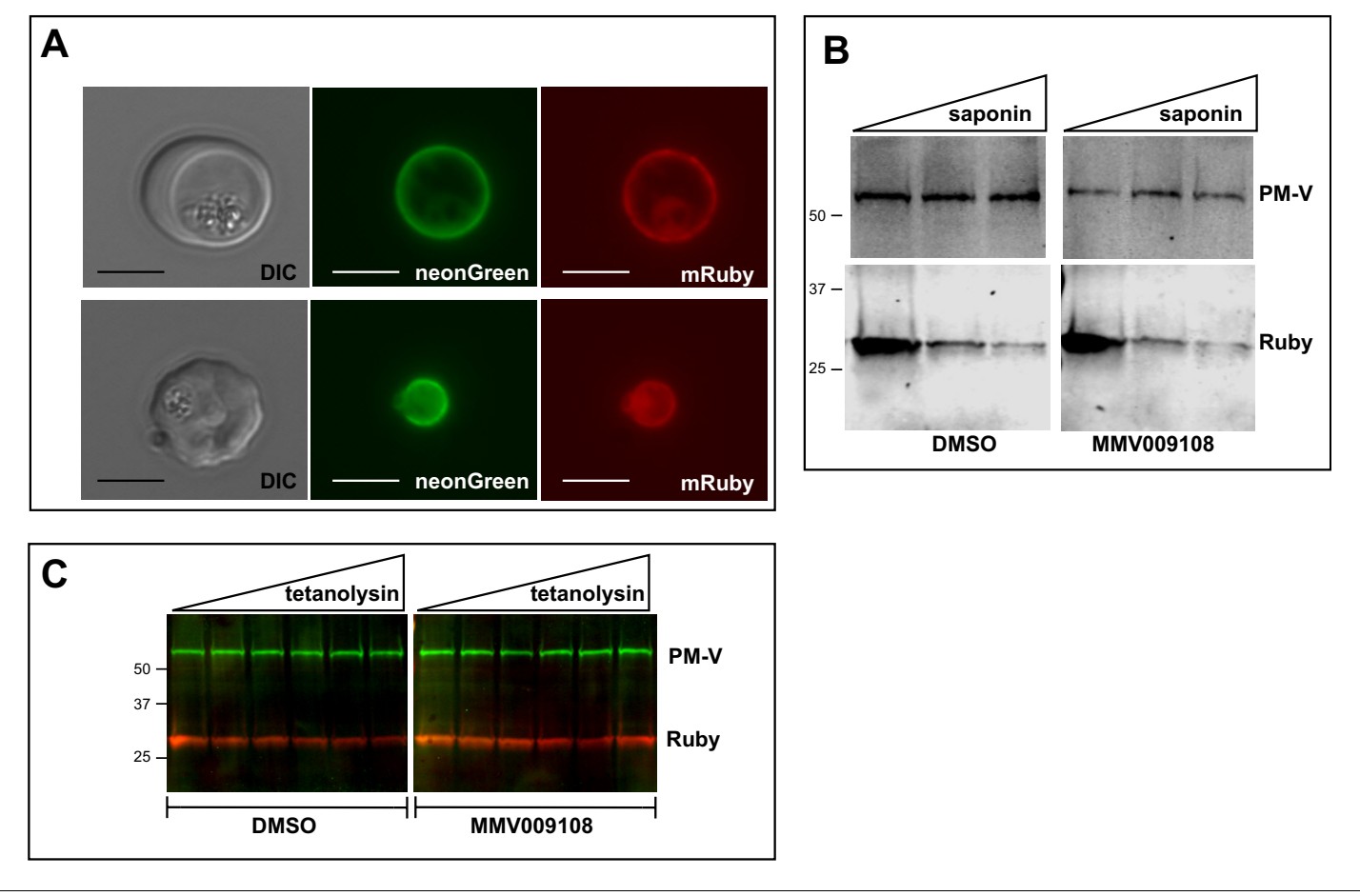

**Figure 5.** PVM lipid homeostasis is not affected by MMV009108. (**A**) Live microscopy on NF54-EXP2-mNeonGreen + PV-mRuby3 parasites. The PVM protein EXP2 is expressed as mNeonGreen fusion; mRuby3 is targeted to PV lumen. Scale bar = 5 μm. (**B**) Western blot on saponin-treated NF54-EXP2-mNeonGreen + PV-mRuby3 parasites following treatment with 500 nM MMV009108 for 2 hr. The saponin gradient was as follows: 0%, 0.009%, 0.018%. This experiment was done two times. (**C**) Western blot on NF54-EXP2-mNeonGreen + PV-mRuby3 parasites following treatment with tetanolysin (concentrations: 0, 0.5, 1, 2.5, 5, 7.5 ng/ml). Blot was probed with anti-RFP and anti-PM-V antibodies. This experiment was done two times. Expected sizes: PV-Ruby3 = 27 kDa, PM-V = 69 kDa.

DOI: https://doi.org/10.7554/eLife.40529.017

The following figure supplement is available for figure 5:

**Figure supplement 1.** Control experiment: PPM but not PVM lipid homeostasis is affected by MMV009108.

DOI: https://doi.org/10.7554/eLife.40529.018

engineered to display endosomal retention signals, ScNCR1 and TgNCR1 were able to revert defective cholesterol transport in mammalian cells lacking functional NPC1 (*Malathi et al., 2004*), though TgNCR1 appears to be selective for sphingomyelin in the parasite. PfNCR1 displays 30% amino acid sequence identity over 69% of TgNCR1. Proof of a direct role of PfNCR1 as a lipid transporter awaits functional analysis. Despite significant homology, there appear to be significant differences as to functions served by the proteins. Whereas ScNCR1 and TgNCR1 are dispensable for survival, PfNCR1 appears to be essential. ScNCR1 has been localized to the yeast vacuole and *T. gondii* NCR1 to the inner membrane complex, a continuous patchwork of flattened vesicular cisternae located beneath the plasma membrane and overlying the cytoskeletal network; PfNCR1 is on the PPM.

Striking phenotypic consequences of PfNCR1 depletion or inhibition provide hints as to the functions served by this transmembrane protein. The ability of the cholesterol-dependent glycoside saponin to release cytosolic content of parasite-infected erythrocytes by permeation of the host

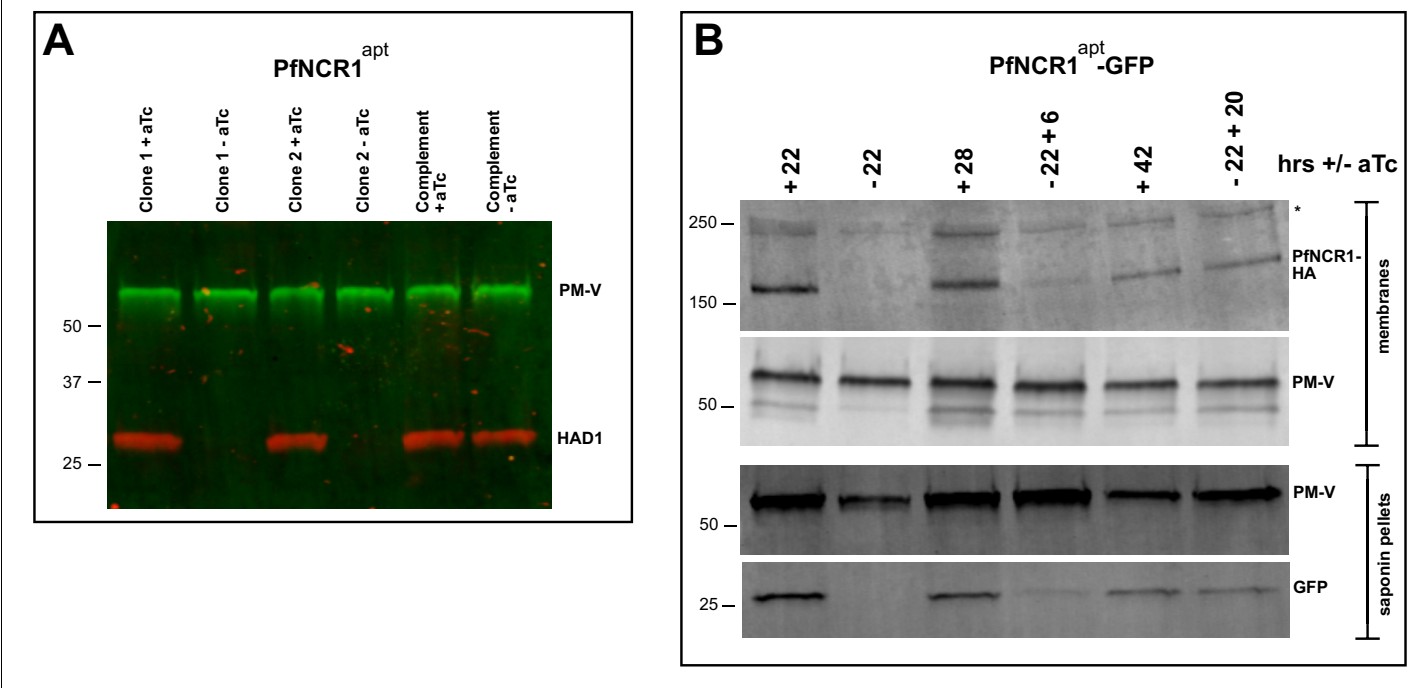

**Figure 6.** PfNCR1 K/D hypersensitizes parasites to saponin. (A) Western blot analysis of saponin extracts (0.07%) from two PfNCR1[apt] clones and complemented parasites. Parasites were harvested 24 hr after aTc washout. This experiment was done two times. (B) Replenishing aTc after washout reverts the K/D phenotype. aTc was removed from PfNCR1[apt]-GFP parasites (stable expression of cytosolic GFP). 22 hr after washout, one set of parasites was harvested, while aTc (500 nM) was added back to another set of parasite samples for 6 or 20 hr. Parasites were either harvested to prepare membranes, or released with saponin. Lysates were subjected to western blotting. * in top blot (anti-HA) marks a cross-reacting protein. This experiment was done two times. Expected sizes: HAD1 = 33 kDa, PM-V = 69 kDa, PfNCR1-HA = 171 kDa, GFP = 27 kDa.

DOI: https://doi.org/10.7554/eLife.40529.019

The following figure supplement is available for figure 6:

**Figure supplement 1.** Saponin hypersensitivity after PfNCR1 K/D is reversed rapidly by addition of aTc.

DOI: https://doi.org/10.7554/eLife.40529.020

plasma membrane while largely sparing the parasite cytosolic content has been a mainstay for experiments requiring 'freeing' of parasites for biochemical and physiological investigations (*Hsiao et al., 1991*). Cholesterol is not synthesized by malaria parasites but is taken up from the erythrocyte and incorporated into parasite membranes. An inward cholesterol gradient is formed as the parasite grows (*Tokumasu et al., 2014*). Resistance of the PPM to saponin permeation is believed to be due to a dearth of cholesterol within the PPM. Furthermore, the accessibility of cholesterol to saponin is highly dependent on its interactions with other lipids (*Aittoniemi et al., 2007*; *Lange et al., 2005*). Interestingly, treatment with PfNCR1-active compounds results in saponin sensitivity of the parasites leading to the release of parasite cytosolic content within a short period of exposure. Remarkably, this saponin sensitivity was reversed upon the removal of the compounds targeting PfNCR1. The reversible saponin sensitivity seen here is reminiscent of effects we have previously reported for antimalarial drugs that inhibit PfATP4, a P-type Na$^+$ pump (*Das et al., 2016*). Induction of saponin sensitivity by PfATP4-active drugs was dependent upon the parasite being grown in a medium with standard [Na$^+$]; saponin sensitivity was not seen in parasites grown in a medium with low [Na$^+$]. Comparing the effects of PfNCR1-active compounds with PfATP4-active compounds, some similarities as well as differences become apparent. Both sets of compounds cause rapid but reversible saponin sensitivity in the PPM. PfATP4-active compounds disrupt Na$^+$ homeostasis, which is a prerequisite for induction of saponin sensitivity, whereas PfNCR1-active compounds induce saponin sensitivity without disrupting Na$^+$ homeostasis (*Figure 4F* and *Figure 4—figure supplement 1C*). It is possible that PfATP4 blockade perturbs the ionic environment critical for PfNCR1 function.

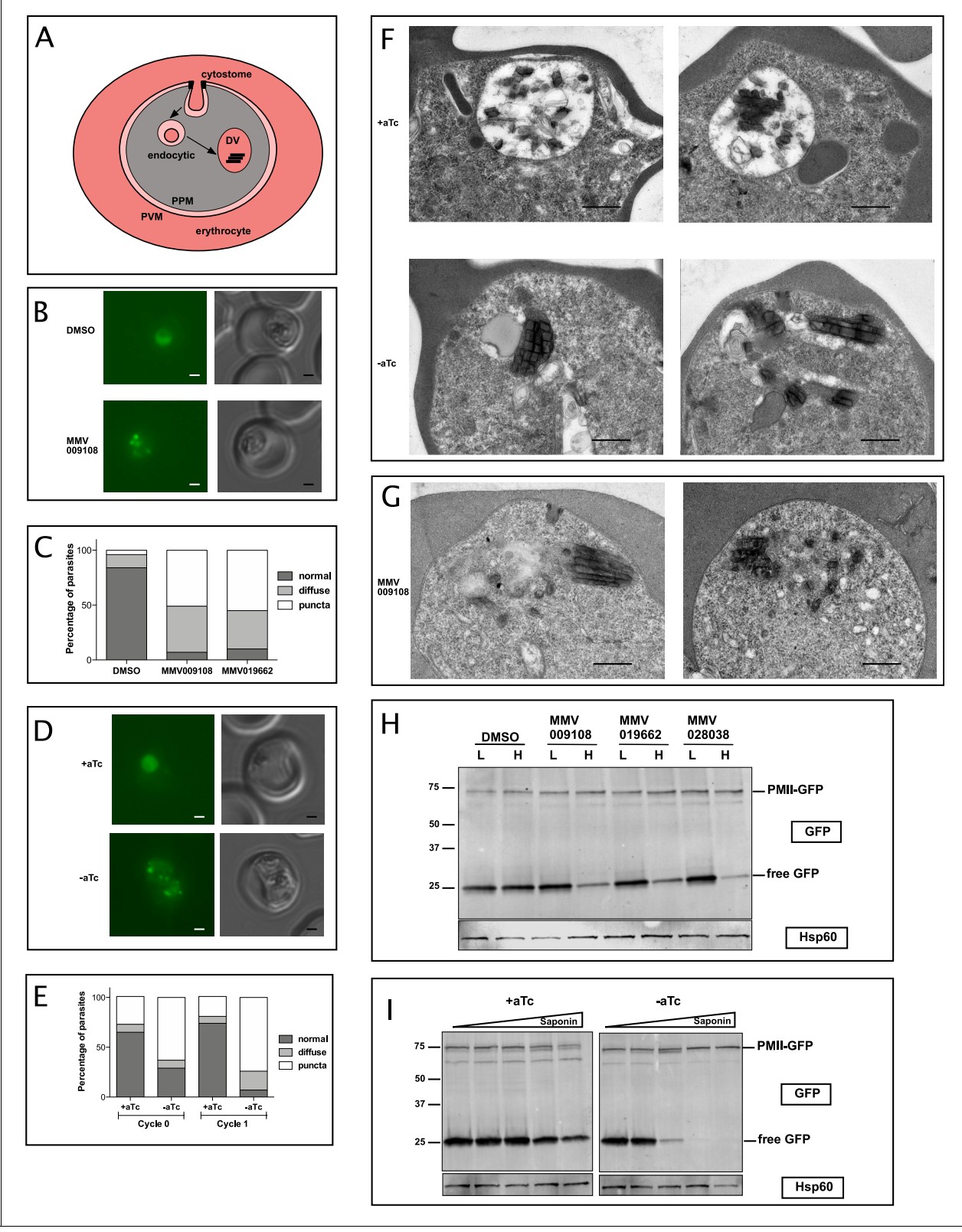

**Figure 7.** PfNCR1 inhibition or K/D impairs digestive vacuole genesis. (**A**) Cartoon of trafficking route to the DV in an infected red blood cell. DV = digestive vacuole; PVM = parasitophorous vacuolar membrane; PPM = parasite plasma membrane. (**B**) Live microscopy of PMII-GFP parasites after incubation with MMV009108 (1 μM, 3 hr) or with vehicle (DMSO). Scale = 1 μm. (**C**) Quantitation of abnormal DVs from parasites in (**B**) after incubation with MMV009108 (N = 43), or MMV019662 (N = 43) or vehicle (DMSO) (N = 77) (1 μM, 3 hr). p<0.0001, Fisher's exact test. (**D**) Live microscopy

*Figure 7 continued on next page*

Figure 7 continued

of PfNCR1 K/D parasites expressing PMII-GFP, after removal of aTc. Scale = 1 μm. (E) Quantitation of abnormal DVs from parasites in (D) after aTc washout. Cycle 0 + aTc (N = 93), –aTc (N = 84); cycle 1 + aTc (N = 107), –aTc (N = 116). p<0.0001, Fisher's exact test. (D and E): cycle 0 = trophozoites after removal of aTc within the same replication cycle (27 hr post washout), cycle 1 = trophozoites after removal of aTc in the preceding replication cycle (68 hr post washout). (F) Transmission electron micrographs of PfNCR1 K/D parasites (clone 2) after aTc removal (68 hr post washout). Scale = 0.5 μm. (G) Transmission electron micrographs of PfNCR1 K/D parasites maintained with aTc and incubated with 500 nM MMV009180 for 1 hr. Scale = 0.5 μm. (H) Western blot analysis of PMII-GFP parasites after treatment with 1 μM compounds for 2 hr. Parasites were released from RBCs with low (L) (0.009%) or high (H) (0.035%) saponin. Top blot was probed with α-GFP antibody, bottom blot (loading control) was probed with α-Hsp60 antibody, an organellar marker. This experiment was done two times. (I) Western blot analysis of PMII-GFP, PfNCR1 K/D parasites after aTc washout for 22 hr. Parasites were released from RBCs with 0.009%, 0.0175%, 0.035%, 0.07% or 0.14% saponin. Top blot was probed with α-GFP antibody, bottom blot (loading control) was probed with α-Hsp60 antibody. This experiment was done two times. Expected size of pro-PMII-GFP=79 kDa, free GFP = 27 kDa.

DOI: https://doi.org/10.7554/eLife.40529.021

The following figure supplements are available for figure 7:

**Figure supplement 1.** Characterization of strain with PfNCR1[apt] in PMII-GFP background.

DOI: https://doi.org/10.7554/eLife.40529.022

**Figure supplement 2.** Examples of different digestive vacuole GFP patterns.

DOI: https://doi.org/10.7554/eLife.40529.023

We noted that the concentrations at which PfNCR1-active compounds caused saponin sensitivity after a short exposure were much lower than the concentrations at which the compounds inhibited parasite growth in 72 hr assays. Similarly, PfNCR1 K/D caused saponin sensitivity of the PPM much sooner than inhibition of parasite growth. These results are opposite of what was previously seen for PfATP4-active compounds (*Das et al., 2016*). Parasites might have a greater tolerance for PPM composition disruption compared to the perturbation of Na$^+$ homeostasis.

Another major consequence of PfNCR1 inhibition or K/D was dramatic changes in formation and morphology of the DVs of parasites. DVs are lysosome-like organelles crucial for degrading hemoglobin. Unlike other eukaryotes or related apicomplexans, malaria parasites must actively digest hemoglobin to create room in the erythrocyte for the growing cell and to generate amino acids for parasite protein synthesis (*Krugliak et al., 2002*; *Rosenthal, 2011*; *Liu et al., 2006*). Uptake of erythrocyte cytosolic contents proceeds via the invagination of the PVM and the PPM and fusion of the PPM with the DV membrane contributes to mature DV formation (*Klemba et al., 2004*). Perhaps the abnormal membrane curvature (*Churchward et al., 2008*) and lack of fusion of the DVs upon loss/inhibition of PfNCR1 function provide clues towards understanding the critical requirement for normal lipid homeostasis in malaria parasites. The accumulation of hemoglobin peptides after incubation with PfNCR1 inhibitors suggests hemoglobin catabolism as a target pathway for these compounds and supports our findings. Among eukaryotes with NCR1 proteins but lacking receptor-mediated sterol uptake, malaria parasites are unusual in their requirement for functional NCR1, thus making this protein an exciting new antimalarial target. The diversity of chemical scaffolds targeting a single critical protein should provide guidance for future drug design and discovery efforts.

## Materials and methods

### Parasite strains, culturing and resistance selection

Parasites were cultured in human red blood cells (2% hematocrit) in RPMI 1640 with 0.25% (w/v) Albumax (cRPMI) as previously described (*Klemba et al., 2004*; *Trager and Jensen, 1976*). A lab-adapted strain of 3D7 that has been fully sequenced was used for most experiments (*Corey et al., 2016*). For GFP overexpression in wild-type parasites, the previously described NF54[eGFP] line was used, which bears an eGFP expression cassette targeted to the *cg6* locus using the attB x attP site-specific integrase recombination system (*Straimer et al., 2012*). Parasites with evolved resistance to MMV009108, MMV028038, or MMV019662 have been described (*Corey et al., 2016*). Briefly, 5 × 10$^8$ to 2 × 10$^9$ 3D7 parasites were pressured with concentrations of 3x-10x EC$_{50}$. Resistant parasites were readily obtained in multiple selections for the three compounds. Resistant and transfected parasites were cloned by limiting dilution. Dose-response experiments were done in triplicate starting with synchronous, young ring-stage cultures (1–1.2% starting parasitemia). Parasitemia (percentage of total erythrocytes infected with parasites) was measured approximately 70–80 hr post compound

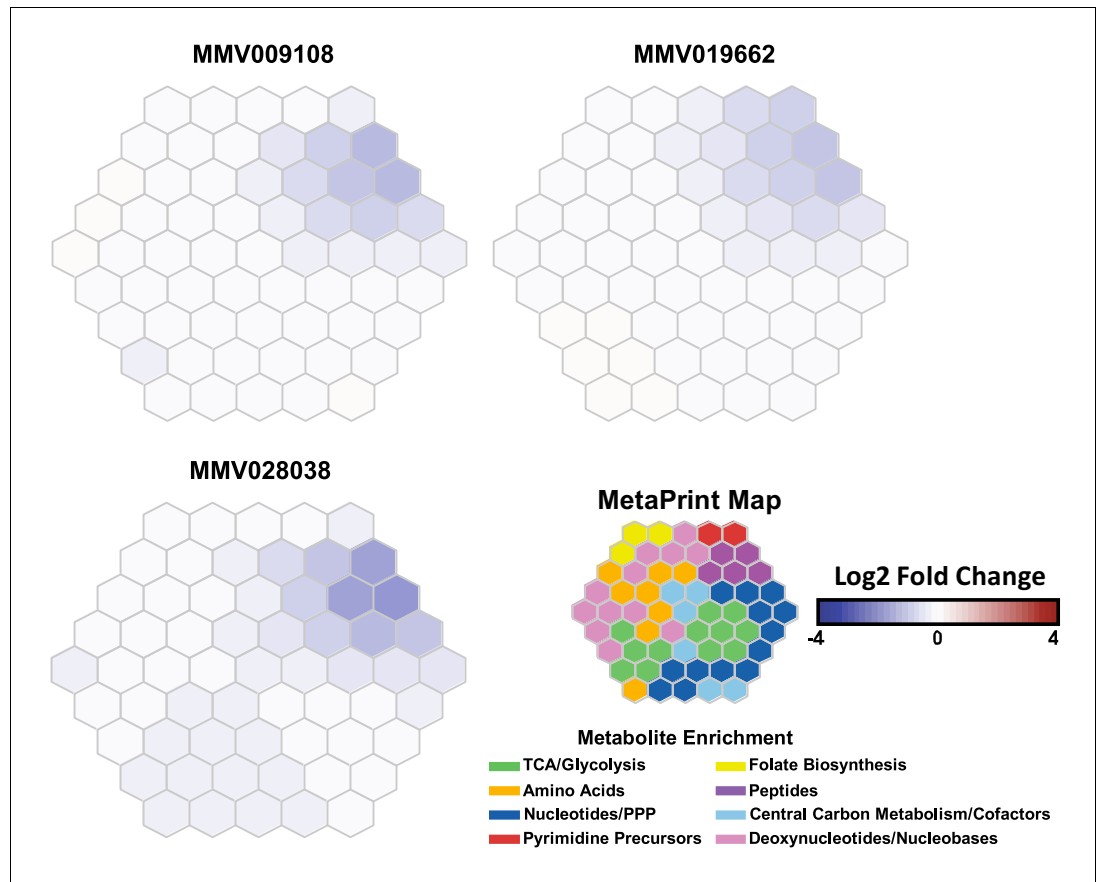

**Figure 8.** Metabolomic analysis of parasites incubated with PfNCR1 inhibitors. Mass spectrometry-based metabolic profiling of hydrophilic extracts from parasites (*Allman et al., 2016*) exposed to the three PfNCR1-targeting MMV compounds depicts a depletion in hemoglobin-derived peptides. Each panel represents incubation with a different compound and is an average of two experiments (each containing triplicates). These Metaprint representations (*Fang and Gough, 2014*) also demonstrate a highly similar metabolic response upon drug treatment with these compounds.

DOI: https://doi.org/10.7554/eLife.40529.024

The following source data is available for figure 8:

**Source data 1.** Log2 fold changes of treated versus untreated controls.

DOI: https://doi.org/10.7554/eLife.40529.025

addition by nucleic acid staining of iRBCs with 0.8 μg/ml acridine orange in PBS. Growth was normalized to parasite cultures with carrier only (DMSO). Chloroquine (500 nM) was used as a positive control for parasite growth inhibition. Data were fit to a sigmoidal growth inhibition curve. Growth curves of K/D and complemented parasites were done in technical triplicate with synchronous parasite cultures (aTc washout at young ring stage) by measuring daily parasitemias. Data were fit to an exponential growth equation. GraphPad Prism 5.0 was used for data analysis. Experiments for monitoring leakage (western blots and flow cytometry) of cytoplasmic HAD1, GFP or mRuby after compound treatment or under PfNCR1 K/D were performed with MACS LD (Miltenyi Biotech, Cat. No. 130-042-901) column-enriched parasites. Parasites were kept in cRPMI during all experiments. For aTc washouts, synchronous young ring-stage parasites were used. Washouts were repeated 3-4x, resuspending parasites at 2% hematocrit in cRPMI with 10 min incubations at room temperature for each washout.

## Saponin release experiments

To monitor sensitivity to saponin, parasite cultures were pelleted (3 min x 840 g), pellets were suspended in 10X volume (most experiments) of room temperate saponin (prepared in PBS) for two

mins (Sigma, Cat. No. S7900). Typical saponin concentration was 0.035%; modifications are indicated in the figure legends of experiments where appropriate. The released parasites were collected by centrifugation (3 min x 2200 g) and washed one time in cold PBS. In the experiment in *Figure 4— figure supplement 1B* (in which both supernatant and pellet fractions were collected) 2X volume saponin was used.

## Tetanolysin release experiments
Magnet-purified synchronous trophozoite-stage parasites were suspended in 10X volume of tetanolysin (0, 0.5, 1, 2.5, 5, 7.5 ng/ml prepared in PBS) and incubated at room temperature for 2 min. The released parasites were collected by centrifugation (3 min x 2200 g) and washed one time in cold PBS.

## Cloning and Southern blots
All plasmids were verified by direct Sanger sequencing. Primers are listed in *Supplementary file 1*.

### Allelic exchange constructs
Allelic exchange constructs were based on the vector pPM2GT (*Klemba et al., 2004*). Basepairs 2305–4893 of PF3D7_0107500 were cloned into the AvrII/XhoI sites using primers AR1-F and AR1-R primers. Using this strategy, *pfncr1* is expressed from the endogenous promoter in-frame with a C-terminal GFP (the native stop is deleted). The mutant constructs were prepared using QuikChange mutagenesis (Agilent Technologies, Cat. No. 20053). For the A1108T mutation, primer Mut-1 was used. In addition to the resistance mutation, this primer also introduces a BspHI site at bp 3294. For the A1208E mutation, primer Mut-2 was used. In addition to the resistance mutation, this primer also introduces a EcoRI site at bp 3605. For the F1436I mutation, primer Mut-3 was used. F1436I mutant parasites also contain a synonymous change at bp4579, resulting in the deletion of a HincII site. 100 μg of circular DNA was transfected by electroporation of ring-stage parasites. Parasites were selected with 5 nM WR99210 (kind gift of D. Jacobus), cycled twice off drug to enrich for parasites with integrated plasmid and cloned by limiting dilution.

### PfNCR1$^{apt}$ parasites
In-Fusion cloning (Clontech) following PCR from gDNA was used to clone right and left homologous regions (RHR and LHR) for integration into the *pfncr1* locus. For the right homologous region, the sequence between bp3671 and bp4893 (the stop was deleted) (primers RHR1F and RHR1R) was amplified. An AflII site was introduced at the 5' end and AatII was introduced at the 3' end. Silent shield mutations to protect the construct from cleavage by CRISPR/Cas9 were introduced at S1464-S1465. For the left homologous region, a 948 bp fragment starting 38 bp past the stop codon was amplified (LHR1F and LHR1R). An AscI site was introduced at the 5' end and an AflII site was introduced at the 3' end. After generation of single homologous region fragments, RHR and LHR PCR products were mixed, amplified with primers RHR1F and LHR1R and cloned into the plasmid pMG75 as described (*Spillman et al., 2017*). The resulting construct (pMG75-PfNCR1) contains a single in frame HA sequence followed by 10x aptamers for aTc-regulatable translational repression. The construct contains two additional amino acids (D,V) before the HA sequence, as two tandem AatII sites were mistakenly introduced. For the gRNA sequence, the sequence 5'-TTAATGTAG TGGGCCAAAAC-3' was chosen. The sense and antisense primer pair GRNA1 and GRNA2 encoding the *pfncr1* sgRNA seed sequence was annealed and inserted into the BtgZI site in plasmid pyAIO (*Spillman et al., 2017*), resulting in the plasmid pyAIO-PfNCR1-gRNA1. 100 μg of pMG75-PfNCR1 was linearized with AflII, purified by phenol-chloroform extraction and co-transfected with 50 μg of pyAIO-PfNCR1-gRNA1 by electroporation. Parasites containing the modified *pfncr1* locus were selected with 5 μg/ml Blasticidin S. For the PfNCR1$^{apt}$ strain that expresses PMII-GFP, we transfected the previously described PMII-GFP clone (*Klemba et al., 2004*) with pyAIO-PfNCR1-gRNA1 and linearized pMG75-PfNCR1. In this case, parasites were selected with 5 nM WR99210 plus 5 μg/ml Blasticidin S and kept in media with 500 nM aTc. Parasites were cloned by limiting dilution.

## Complementation of PfNCR1[apt]

For complementation, RNA was prepared from 3D7 parasites using TRIzol (ThermoFisher), *pfncr1* RNA was amplified using a SuperScript RT-PCR kit (Invitrogen) with primers Comp1 and Comp2, cloned into the XhoI/AvrII sites of the pTEOE random integration vector with the PiggyBac transposase as described (*Sigala et al., 2015*; *Balu et al., 2005*). PfNCR1[apt] clone two was transfected and selected with 5 μg/ml Blasticidin S and 2 μM DSM-1 (Asinex) (*Ganesan et al., 2011*).

## Expression of cytoplasmic GFP in PfNCR1[apt] background

GFP overexpression in PfNCR1[apt] parasites was achieved by targeting the eGFP expression cassette of NF54[eGFP] parasites to the *rh3* locus by CRISPR/Cas9 editing. The *calmodulin* promoter and *egfp* coding sequencing was amplified from NF54[eGFP] genomic DNA template using primers eGFP-F and eGFP-R and inserted into the plasmid pPM2GT (*Klemba et al., 2004*) between AatII and EagI by In-Fusion cloning, allowing for fusion to the *hsp86* 3' UTR. The sgRNA target site TGGTAATACAGAAA TGGATG was chosen in the dispensable *rh3* gene. Homology flanks were then amplified from sequence just upstream and downstream of the Cas9 cleavage site defined by this sgRNA using primers Rh3-5'F/R and Rh3-3'F/R. These amplified flanks were used as template and assembled into a single DNA molecule with an intervening AflII site in a second PCR reaction using primers Rh3-5'R and Rh3-3'F and this flank assembly was inserted into the BglII site of the pPM2GT-CAM-eGFP plasmid resulting in the plasmid pPM2GT-CAM-eGFP-RH3-flanks. A sense and antisense primer pair (Rh3-G1 and Rh3-G2) encoding the *rh3* sgRNA seed sequence was annealed and inserted into the BtgZI site in plasmid pyAIO (*Spillman et al., 2017*) resulting in the plasmid pyAIO-RH3-gRNA1. Plasmid pPM2GT-CAM-eGFP-RH3-flanks was linearized at AflII and co-transfected with pyAIO-RH3-gRNA1 into PfNCR1[apt] clone two and selected with 2 μM DSM-1 for integration into the rh3 locus.

## Expression of split-GFP

For split GFP experiments, two parasite lines were generated expressing either PV-targeted or cytosolic GFP1-10. A fusion of the *sera5* signal peptide and *gfp1-10* coding sequence was synthesized as a gBlock (gBlock1; IDT) and used as template to PCR amplify *gfp1-10* with primers GFP1-10-1F and GFP1-10-1R or without the *sera5* signal peptide (primers GFP1-10-2F and GFP1-10-1R). These amplicons were inserted into plasmid pLN-ENR-GFP (*Adjalley et al., 2010*) between AvrII and AflII to generate plasmids pLN-SP-GFP1-10 and pLN-GFP1-10, respectively. Each plasmid was co-transfected with plasmid pINT into NF54[attB] parasites and selected with 2.5 μg/ml Blasticidin S to facilitate integration into the *cg6* locus through integrase-mediated attB x attP recombination (*Adjalley et al., 2010*). A clonal line was derived from each transfected parasite population by limiting dilution and designated NF54[pvGFP1-10] or NF54[cytGFP1-10], respectively. GFP1-10 expression and targeting to the proper compartment (parasitophorous vacuole or cytosol) was confirmed by western blot and immunofluorescence assay using a rabbit-anti-GFP (Abcam 6556). For endogenous tagging of PfNCR1 with 3xHA-GFP11, *pfncr1* was amplified from pMG75-PfNCR1 with primers GFP11-F and GFP11-R and inserted into the plasmid pyPM2GT-EXP2-mNeonGreen (*Glushakova et al., 2017*) between XhoI and AvrII. Transfections were selected with 2 μM DSM-1. This construct expresses PfNCR1-GFP11 from its native promoter.

## For monitoring PVM integrity

The line NF54-EXP2-mNeonGreen + PV-mRuby3 was used (*Glushakova et al., 2018*).

## Southern blot

To confirm correct integration, we used the AlkPhos Direct Kit (FisherScientific Cat. No. 45-000-936) for Southern blots as described (*Klemba et al., 2004*). For the probe, we amplified a 674 bp fragment from gDNA using primers Probe1 and Probe2.

## Microscopy

Fluorescence microscopy was performed on live, GFP-expressing parasites using a Zeiss Axioskope. Nucleic acid was detected by staining with DAPI. For *Figures 3D–G* and *Figure 7B and D*, background correction was done using the program Affinity Designer and was applied consistently for all figures. For *Figure 3D*, spinning-disc confocal images of live or immunolabeled cells were captured

and analyzed on an AxioObserver Z1 (Carl Zeiss, Inc) with a 60X oil objective, running Zen two software (Carl Zeiss, Inc).

For electron microscopy, infected RBCs were enriched using MACs LD columns, fixed in 4% paraformaldehyde (Polysciences Inc, Warrington, PA) in 100 mM PIPES/0.5 mM MgCl2, pH 7.2 for 1 hr at 4°C. Samples were then embedded in 10% gelatin and infiltrated overnight with 2.3M sucrose/20% polyvinyl pyrrolidone in PIPES/MgCl2 at 4°C. Samples were trimmed, frozen in liquid nitrogen, and sectioned with a Leica Ultracut UCT cryo-ultramicrotome (Leica Microsystems Inc, Bannockburn, IL). 50 nm sections were blocked with 5% FBS/5% NGS for 30 min and subsequently incubated with rabbit anti-GFP (Life Technologies; Cat. No. A11122) (1:500) for 1 hr, followed by goat anti-rabbit IgG (H + L) antibody conjugated to 18 nm colloidal gold (1:30) (Jackson ImmunoResearch) for 1 hr. Sections were washed in PIPES buffer followed by a water rinse, and stained with 0.3% uranyl acetate/ 2% methyl cellulose and viewed on a JEOL 1200EX transmission electron microscope (JEOL USA, Peabody, MA) equipped with an AMT eight megapixel digital camera (Advanced Microscopy Techniques, Woburn, MA). All labeling experiments were conducted in parallel with controls omitting the primary antibody which was consistently negative at the concentration of colloidal gold conjugated secondary antibodies used in these studies. For EM without immunostaining, cells were fixed in 2% paraformaldehyde/2.5% glutaraldehyde (Polysciences Inc, Warrington, PA) in 100 mM sodium cacodylate buffer, pH 7.2 for 1 hr at room temperature. Samples were washed in sodium cacodylate buffer and postfixed in 1% osmium tetroxide (Polysciences Inc) for 1 hr. Samples were then rinsed extensively in dH2O prior to en bloc staining with 1% aqueous uranyl acetate (Ted Pella Inc, Redding, CA) for 1 hr. Following several rinses in dH2O, samples were dehydrated in a graded series of ethanol and embedded in Eponate 12 resin (Ted Pella Inc). Sections of 95 nm were cut with a Leica Ultracut UCT ultramicrotome (Leica Microsystems Inc, Bannockburn, IL), stained with uranyl acetate and lead citrate, and viewed on a JEOL 1200 EX transmission electron microscope (JEOL USA Inc, Peabody, MA) equipped with an AMT eight megapixel digital camera and AMT Image Capture Engine V602 software (Advanced Microscopy Techniques, Woburn, MA).

## Flow cytometry

For flow cytometry experiments with eGFP, 50,000 cells were counted on a BD FACSCanto and scored for high or low GFP signal. Appropriate gating of cells was established using untreated parental or NF54$^{eGFP}$ parasites.

## Western blotting

For PfNCR1 blots, membrane preparations were made. $1 \times 10^8$ to $5 \times 10^8$ trophozoite-stage parasites were released from RBC with 0.035% saponin, washed in cold PBS, resuspended in 300 µl DI-water with protease inhibitors (HALT, ThermoFisher, Cat. No. 78430), freeze-thawed 3x with liquid nitrogen/42°C water bath. The membranes were pelleted (17 k g), resuspended in 100 µl-300µl (depending on sample amount) Ripa buffer (25 mM Tris (pH 7.6), 150 mM NaCl, 1% NP-50, 0.1% SDS, 1% Sodium Deoxycholate) containing 0.1% CHAPS and 0.1% ASB-14, sonicated 3x with a microtip, and incubated at 42°C with shaking for 45 min. The samples were then centrifuged (17 k g, 30 min), SDS sample buffer was added to the soluble portions. The samples were warmed at 42°C and loaded on 4–15% TGX gradient gels (Biorad). Proteins were transferred onto PVDF using wet transfer with 20% methanol. Blots were blocked either 1 hr at 25°C or overnight at 4°C with Licor Odessey block buffer. Primary antibodies were mouse monoclonal α-HA antibody (Biolegend) at 1:1000 or LivingColors mouse-α-GFP (Takara, Cat. No. 632380) (1:1000). For the loading control mouse monoclonal α-PM-V antibody (*Banerjee et al., 2002*) at 1:20 was used. Secondary antibody was goat-α-mouse (800) IR-Dyes (1:20,000) from Licor.

For western blot monitoring leakage of cytosolic proteins after incubation with compound or PfNCR1 K/D, parasites were resuspended in saponin-containing PBS, pelleted, lysed in Ripa buffer containing protease inhibitors and with brief sonication. Soluble proteins after centrifugation (30 min, 17 k g) were added to sample buffer, briefly heated at 98$^0$C and loaded onto 10% or 12% TGX gels (Biorad). Western blotting was done using the protocol indicated above. Primary antibodies were: rabbit α-HAD1 (a gift from Dr. Audrey Odom John, WU) (*Guggisberg et al., 2014*) (1:1000), rabbit α-Hsp60 (1:500) (a gift from Dr. Sabine Rospert, University of Freiburg), mouse α-PM-V (1:20) (*Banerjee et al., 2002*), rabbit α-RFP (1:1000) (Thermofisher, Cat. No. R10367), mouse α-GFP (Living

Colors JL-8, Clontech, Cat. No. 632380) (1:1000), HRP-conjugated α-aldolase (Abcam, Cat. No. ab38905) (1:10000), α-EXP2 antibody (gift from Professor James Burns, Drexel University) (1:10000) (*Das et al., 2016*). Secondary antibodies were goat-α-mouse (800) and donkey-α-rabbit (680) IR-Dyes (1:20,000) from Licor. Immunoblots shown in *Figure 4A and E* and *Figure 6—figure supplement 1* were washed in PBS-Tween (0.2%) and developed using the Super Signal West Pico Chemi-luminescent substrate (Thermo Scientific, Cat. No. 34080).

## Measuring intracellular [Na$^+$]

Intracellular Na$^+$ measurements for parasites were performed using methods adapted from *Spillman et al. (2013)*. Briefly, *P. falciparum* cultures were loaded with the sodium-sensitive dye SFBI-AM (5.5 μM) (Molecular Probes) and 0.02% w/v Pluronic F-127 (Molecular Probes) in RPMI at 37°C for 1 hr. Loaded parasite cultures were diluted to 5% hematocrit and freed from host red blood cells by exposing the culture to 0.05% w/v saponin (Sigma-Aldrich #47036) for 15–20 s and pelleted by centrifuging at 500x g, 5 min. Freed parasites were washed twice (2000x g, 30 s) and resuspended to a final concentration of 5–7.5 × 10$^7$ cells/mL in a saline buffer (125 mM NaCl, 5 mM KCl, 1 mM MgCl$_2$, 20 mM glucose, 25 mM HEPES, pH 7.3). SBFI-loaded parasites were excited at 340 nm and 380 nm with emissions recorded at 505 nm at 37°C in a fluorescence spectrophotometer (Hitachi F-7000). Auto-fluorescence corrected SBFI emissions at 340 nm and 380 nm were plotted as ratios.

## Metabolomic profiling

Changes in metabolites were measured in response to compounds using whole cell hydrophilic extraction, followed by ultra-high precision liquid chromatography mass-spectrometry (UHPLC-MS) using negative ionization as in Cowell et al., 2018 (*Cowell et al., 2018*). This was performed on synchronous, trophozoite infected red blood cells (iRBCs, 24–36 hpi) which had been magnetically separated from culture. Quantification of cells was performed by hemocytometry, and treatments were performed on 1 × 10$^8$ iRBCs in wells containing 5 mL of RPMI. Treatment conditions were performed in triplicate, with compound concentrations of 10xEC$_{50}$ for 2.5 hr, followed by washing with PBS and extraction using 90% methanol containing isotopically-labeled aspartate as an internal standard for sample volume. Samples were dried using nitrogen prior to resuspension in water containing 0.5 uM chlorpropamide as an internal standard for injection volume. Samples were then analyzed via UHPLC-MS on a Thermo Scientific EXACTIVE PLUS Orbitrap instrument as established in Allman et al., 2016 (*Allman et al., 2016*).

## Acknowledgements

We are thankful to our MalDA Consortium collaborators and DS Ory (WU) for stimulating discussions, AS Nasamu and A Polino for valuable suggestions, B Vaupel for assistance during cloning, W Beatty for electron microscopy, LD Sibley for use of the spinning-disk confocal microscope and M Lee, M Carrasquilla and J Rayner for consulting on the Rh3 gRNA design. This work was supported by Gates Foundation Grants OPP 1054480 (Winzeler, Goldberg, Llinás), OPP1132313 (Niles), and NIH grants R01AI098413 and R01AI132508 (Vaidya); K99/R00 HL133453 (Beck), and 1DP2OD007124 (Niles).

## Additional information

### Funding

| Funder | Grant reference number | Author |
|---|---|---|
| Bill and Melinda Gates Foundation | OPP 1054480 | Eva Istvan<br>Edward Owen<br>Manuel Llinas<br>Elizabeth Winzeler<br>Daniel E Goldberg |
| National Institute of Allergy and Infectious Diseases | R01AI132508 | Sudipta Das<br>Suyash Bhatnagar<br>Akhil B Vaidya |

| National Heart, Lung, and Blood Institute | K99/R00 HL133453 | Josh R Beck |
| Bill and Melinda Gates Foundation | OPP1132313 | Suresh M Ganesan<br>Jacquin C Niles |
| National Institutes of Health | 1DP2OD007124 | Jacquin C Niles |
| National Institute of Allergy and Infectious Diseases | R01AI098413 | Akhil B Vaidya |

The funders had no role in study design, data collection and interpretation, or the decision to submit the work for publication.

### Author contributions
Eva S Istvan, Sudipta Das, Conceptualization, Data curation, Formal analysis, Validation, Investigation, Visualization, Methodology, Writing—original draft, Writing—review and editing; Suyash Bhatnagar, Data curation, Formal analysis, Validation, Visualization, Methodology; Josh R Beck, Conceptualization, Resources, Methodology, Writing—review and editing; Edward Owen, Data curation, Formal analysis, Investigation, Visualization; Manuel Llinas, Resources, Data curation, Formal analysis, Investigation, Writing—review and editing; Suresh M Ganesan, Resources, Formal analysis, Investigation, Writing—review and editing; Jacquin C Niles, Resources, Formal analysis, Supervision, Funding acquisition, Writing—review and editing; Elizabeth Winzeler, Resources, Supervision, Funding acquisition, Project administration, Writing—review and editing; Akhil B Vaidya, Daniel E Goldberg, Conceptualization, Resources, Supervision, Funding acquisition, Validation, Investigation, Methodology, Writing—original draft, Project administration, Writing—review and editing

### Author ORCIDs
Eva S Istvan (ID) https://orcid.org/0000-0002-8666-3248
Josh R Beck (ID) https://orcid.org/0000-0001-6196-8689
Manuel Llinas (ID) http://orcid.org/0000-0002-6173-5882
Elizabeth Winzeler (ID) http://orcid.org/0000-0002-4049-2113
Akhil B Vaidya (ID) http://orcid.org/0000-0003-1063-5571
Daniel E Goldberg (ID) http://orcid.org/0000-0003-3529-8399

### Decision letter and Author response
Decision letter https://doi.org/10.7554/eLife.40529.031
Author response https://doi.org/10.7554/eLife.40529.032

# Additional files

### Supplementary files
• Supplementary file 1. Primers used in this manuscript.
DOI: https://doi.org/10.7554/eLife.40529.026
• Transparent reporting form
DOI: https://doi.org/10.7554/eLife.40529.027

### Data availability
All source data are included in the manuscript. Complete metabolomic data has been deposited at Metabolomics Workbench (doi: 10.21228/M8DH49).

The following dataset was generated:

| Author(s) | Year | Dataset title | Dataset URL | Database and Identifier |
| --- | --- | --- | --- | --- |
| Eva Istvan, Sudipta Das, Suyash Bhatnagar, Josh R Beck | 2019 | Metabolomic Data from: '*Plasmodium* Niemann-Pick type C1-related protein is a druggable target required for parasite membrane homeostasis' | http://doi.org/10.21228/M8DH49 | UCSD Metabolomics Workbench, 10.21228/M8DH49 |

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
