## [Decision Letter]

[**Editorial note:** This article has been through an editorial process in which the authors decide how to respond to the issues raised during peer review. The Reviewing Editor's assessment is that all the issues have been addressed.]

Thank you for submitting your article "*Plasmodium falciparum* Niemann-Pick Type C1-Related Protein:a Druggable Target Required for Parasite Membrane Homeostasis" for consideration by *eLife*. Your article has been reviewed by three peer reviewers, and the evaluation has been overseen by a Reviewing Editor and Anna Akhmanova as the Senior Editor. The reviewers have opted to remain anonymous.

The Reviewing Editor has highlighted the concerns that require revision and/or responses, and we have included the separate reviews below for your consideration. If you have any questions, please do not hesitate to contact us.

Istvan and colleagues functionally characterized a putative lipid transporter named PfNCR1 based on some homology to a human Neimann-Pick C1-related cholesterol transporter. They identified this gene through in vitro selection of resistant mutants with 3 structurally distinct lead compounds in whole-cell antimalarial growth inhibition screens. Allelic exchanges to introduce the selected mutations increased IC50 values for the antimalarial compounds thus supporting a role of the gene product in resistance. Conditional knockdown of the transporter also increased sensitivity to the antimalarial compounds. The lack of expression or chemical inhibition of the protein results in an increased sensitivity of the parasite plasma membrane and digestive vacuole membrane to saponin, an amphipathic glycoside with a high affinity for cholesterol. The authors propose that PfNCR1 serves a role in cholesterol export and localizes to the parasite plasma membrane using split GFP constructs, in contrast to a lysosomal localization for the human ortholog. Taken together the data are consistent with PfNCR1 having a role in regulating the lipid composition of the plasma membrane and digestive vacuole membrane and qualify as a a novel *Plasmodium falciparum* candidate drug target.

As noted in the reviews below, the reviewers agree that the findings are of broad interest and suitable novelty for presentation in *eLife*. However, they identified major weaknesses and expect that you will be able to address their various concerns described below by revising the work.

All of the reviewers have raised serious concerns about statistical significance as it relates to several critical findings. In addition, the interpretation of NCR1 localization and its presumed role and essentiality are insufficiently supported by the data presented.

Separate reviews (please respond to each point):

*Reviewer #1*

This well written manuscript reports on a combination of both chemical compounds and genetics to identify PfNCR1 as a novel *Plasmodium falciparum* drug target. The authors used three chemically diverse compounds for selections to induce resistance in a wild type parasite line and confirmed the resulting single nucleotide polymorphisms (SNPs) as the drivers of resistance by introducing them in wild type backgrounds via single-crossover allelic exchange. Two of the mutations, A1108T and F1436I, were found to confer resistance to all three compounds while A1208E generated resistance only to the selecting compound. An effort to disrupt pfncr1 proved unsuccessful, suggesting essentiality. This was further substantiated by a genetic knock down (K/D) using a regulated aptamer translation repression technology showing that PfNCR1 is important for asexual parasite viability. This was further validated by the rescue of K/D growth defects by complementing wild type PfNCR1. Using split GFP constructs, the authors were able to show that PfNCR1 is localized on the parasite's plasma membrane (PPM), in contrast to an earlier study that suggested digestive vacuole (DV) localization. On the basis of known or predicted functions of human, yeast or Toxoplasma orthologs in cholesterol transport, the authors assessed the effects of saponin, an amphipathic glycoside with a high affinity for cholesterol, on compound treated or PfNCR1 K/D parasites. The results showed that in both cases, the parasites were hyper-sensitized to saponin, suggesting that 1) the tested compounds interfered with PfNCR1 activity and 2) that PfNCR1 function is required for the maintenance of PPM lipid composition. Because parasite DVs are formed from vesicles originating from the PPM, the authors investigated whether PfNCR1 activity is also required for DV function. Their results provided evidence that both compound-treated and PfNCR1 K/D parasites had deformed DVs, validating their initial hypothesis.

This is a focused and well executed study. I found the manuscript to be clear and well-written and deserving of publication in *eLife*. Listed below are several suggestions that could benefit the manuscript.

1) I could not find the IC50s of the different compounds. Please add these to the manuscript or to a supplementary table.

2) Many the experiments reported in this manuscript were done in biological duplicates (N = 2). This is not sufficient to claim statistical significance, at least not if the data were part of a major figure in the paper. A third repeat would be desirable (Figure 1 E-F) and might even help data interpretation regarding Figure 4 B-D. Parasite leakiness in response to saponin and MMV009108 seems lower at 50 and 100 nM of compound versus that observed when treated with MMV028038 and MMV019662. Can this effect be quantified as IC50s to allow for easier comparison between different treatments? Is the lower effect of MMV009108 related to a lower general activity of MMV009108 versus the other compounds?

3) The GFP signal in Figures 3 and 7 is either too diffuse or is not sufficient in labeling organelles of interest in the microscopy images shown, without the presence of other parasite organelle markers. I would propose to either perform co-staining with known organelle markers to better orient the reader, or move the live-cell and fixed IFA images to the supplement. The immuno-electron microscopy images provided are clearer and by themselves appear satisfactory in showing localization and DV deformation.

Minor:

1) Figure 1C – Could you elaborate on how the schematic was generated based on the limited sequence similarity between hNPC1 and PfNCR1? Was this achieved via visual association?

2) Figure 2—figure supplement 1A – A schematic of the plasmid, the wild type locus and the subsequent recombinant locus would be more informative when interpreting the gel.

3) Figure 2A – The legend says that the harvested parasites were trophozoite stage yet the text in subsection "PfNCR1 is Important for Asexual Parasite Viability and is Targeted by Antimalarial Compounds" of the Results section says they were highly synchronized young rings. Please clarify.

4) Figure 2C – Is there any obvious explanation as to why the complemented strain proliferates less well than the PfNCR1apt with aTc, especially since the legend mentions that the rescue was done using a stably expressed wt PfNCR1 strain? Also, please report the missing R2 in the legend part C for "+aTc".

5) Figure 2 D-H – A key beside the panels would be helpful.

6) The different subpanels of Figure 4 B-D are mislabeled in the figure legend.

*Reviewer #2*

This is a very interesting study in which an uncharacterised transport protein in *P. falciparum* was shown to be essential for normal growth, implicated as the target of three structurally diverse chemicals, and localised to the parasite plasma membrane. Multiple lines of evidence were provided that the lack of expression or chemical inhibition of the protein results in an increased saponin sensitivity of the parasite plasma membrane and digestive vacuole membrane. The data are consistent with the protein having a role in regulating the lipid composition of the plasma membrane and consequently the DV membrane.

One concern is that straightforward experiments like FACS-based parasite growth assays (Figure 1 D-F, Figure 2 D,F-H) and measurements of GFP release (Figure 4 C-D) were performed fewer than three times, with data from a single experiment shown. It is not stated how many times the Na^+^ experiment shown in Figure 4F, or the growth assays shown in Figure 1—figure supplement 2 and Figure 4—figure supplement 1C, were performed. Experiments such as these should be performed at least three times and averaged data should be shown or averaged IC50/EC50/rate values provided. The low number of biological replicates means that statistical comparisons cannot be (and were not) performed for these data. Very few of the findings in this study (only the data in Figure 7 C,E) are verified as being statistically significant. The authors state that the PfNCR1 K/D parasites are 'minimally hypersensitive' to KAE609. This statement is not justified based on a subtle difference shown in a single experiment.

1) It is not clear how the experiments shown in Figure 2 B and C were performed or how the cumulative parasitemia was calculated. What was the starting parasitemia? Were the cultures diluted and supplemented with uninfected RBCs when required?

2) Figure 7B and 7E – it would be good to see a few representative images for each treatment rather than just one so that readers can understand what was categorised as 'diffuse' and what was categorised as 'puncta'.

3) It is not clear what the authors mean when they say that the Na^+^ measurements were calibrated 'using an average from 3 independent calibration curves for SBFI'. Do they mean that three sets of calibrations were performed within each experiment? Or were calibration curves obtained in three initial experiments used to calibrate data obtained in subsequent experiments? Given that the precise cell number and cell age distribution (as well as other factors) will affect the fluorescence ratio obtained in a particular experiment, it would not be appropriate to apply calibration data that were obtained in a different experiment.

4) Can the authors comment (in the manuscript) on how the resistance levels of the transfectant lines compare to the resistance levels of the compound-selected parasites?

5) Can the authors explain (in the manuscript) the rationale for engineering only one of the two mutations selected for by MMV028038 and MMV019662?

6) Can the authors comment (in the manuscript) on whether their inability to knock out pfncr1 was consistent with the results obtained in the mutagenesis study by Zhang et al. (Science)?

7) Is it surprising that the PfNCR1 knock-down parasites grew as well as they did? Have the authors attempted to quantify how much PfNCR1 expression still occurs in the absence of aTc?

8) Figure 1 title needs correcting: 'Mutations in PfNCR1 confer resistance mutations…'

9) Figure 1 legend needs correcting: 'resistant mutations'

10) Figure 4 legend needs correcting: 'Treated parasites were washed out…'

11) Some of the panel letters referred to in the Figure 4 legend are not used in the correct place.

12) What does 'SP' refer to in Figure 3? Would 'PV' be more appropriate?

13) Figure 6 – can HA on the figure be replaced with PfNCR1-HA?

There was a lack of statistical information because many experiments were not performed enough times.

*Reviewer #3*

Istvan and colleagues report molecular and cell biology studies of a putative lipid transporter (PF3D7_0107500) which they rename PfNCR1 based on some homology to a human Neimann-Pick C1-related cholesterol transporter. The authors identified this gene through in vitro selection of resistant mutants with 3 lead compounds in whole-cell antimalarial growth inhibition screens. Allelic exchange to introduce the selected mutations increases IC50 values for the antimalarial compounds, supporting a role of the gene product in resistance. Conditional knockdown of the transporter also increased sensitivity to the antimalarial compounds. The authors suggest that PfNCR1 is essential, serves a role in cholesterol export and localizes to the parasite plasma membrane, in contrast to a lysosomal localization for the human ortholog.

1) Localization of the putative transporter: The authors show that the GFP-tagged PfNCR1 produces a fluorescence signal around the intracellular parasite (Figure 3A). Immuno-EM using an anti-GFP antibody supports this with gold-dot labeling of either the parasite plasma membrane (PPM) or the PVM; as immuno-EM could not distinguish between the PPM and PVM, they used an interesting split-GFP strategy with expression of soluble GFP1-10 and a short GFP11 tag at the C-terminus of PfNCR1. When GFP1-10 was targeted to the parasite cytosol, they observed a diffuse fluorescence signal in the parasite cytosol (Figure 3D); when GFP1-10 is targeted to the lumen of the PV, no signal was observed. They interpret this as supporting a PPM localization with the C-terminus projecting into the parasite cytoplasm. This approach, though creative and well-executed, raises two concerns.

a) The results equally well support a PVM localization with the protein C-terminus projecting into host cytosol. In this alternate model, the signal observed with GFP1-10 in parasite cytosol would represent protein in transit from the parasite ER and exported via vesicular trafficking. Although it is not clear how parasite-derived membrane proteins reach the PVM, export from the PPM on exocytic vesicles and subsequent fusion with the PVM is a favorite model of most workers. Because the GFP11 tag would be intravesicular during this transit and then exposed to host cytosol after insertion, a fluorescence signal with GFP1-10 in the PV lumen would not be expected. Thus, the split-GFP strategy used cannot unambiguously distinguish between PPM and PVM localizations. The authors' statement "our model suggests that the relative orientation of the cytosolic regions in these two distantly related proteins is conserved" applies equally well to both the PPM and PVM models, and so does not help resolve. The authors should consider indirect immunofluorescence after erythrocyte membrane permeabilization with tetanolysin to explore a PVM localization, as has been used by other workers (JBC 278:48413, 2003); while a positive result would conclusively support a PVM localization, a negative result would warrant further experimentation.

b) The fluorescence signals shown in Figure 3A (full-length GFP tag on PfNCR1) and Figure 3D (short GFP11 tag on PfNCR1) appear to be noticeably different from one another, with a rim pattern for the full-length GFP tag and a more diffuse cytoplasmic pattern for the short GFP11 tag. This raises the possibility that a large, full-length GFP tag may cause PfNCR1 mis-trafficking to the PPM/PVM. Did the authors attempt IFA with anti-HA and the PfNCR1-apt parasite, which has an acceptably small epitope tag? This reviewer feels this is necessary, in part because the PPM localization contrasts with the more conventional localization of NCR1 to the lysosome in other organisms; the abnormal DV phenotype in the PfNCR1 knockdown could also suggest that the wild-type protein localizes to the DV (the parasite lysosome-equivalent).

2) Presumed transport activity: The authors name the putative transporter NCR1 based on weak homology to the human NCR1 and orthologs in Saccharomyces and Toxoplasma; I would caution against assigning a name and role based on computational analysis alone. A role in cholesterol transport for ScNCR1 and TgNCR1 were supported by complementation experiments in mammalian cells lacking NPC1. The authors present an increased saponin sensitivity of the parasite upon PfNCR1 knockdown or inhibitor treatment as evidence for cholesterol transport, but they made the same observations in PfATP4 studies in a prior publication. Why didn't they propose a role in cholesterol transport for that transporter? A more cautious view might be that increased saponin sensitivity is a nonspecific parasite response to induced defects at the PPM. Complementation experiments such as performed for ScNCR1 and TgNCR1 appear to be critical for supporting their model. Control knockdowns followed by saponin-sensitivity experiments for other membrane proteins at the PPM may also help to exclude a nonspecific parasite response.

3) Essentiality: The authors emphasize that, in contrast to presumed orthologs in other organisms, PfNCR1 is essential. This seems to be based on negative results with attempted CRISPR knockout and on a delayed slow growth phenotype with conditional knockdown using the TetR-DOZI/aptamer system. Remarkably, the delayed slow growth phenotype appears to take ~ 10 days (about 5 asexual cycles) to be manifest clearly as no statistics on differences in growth rate for plus and minus aTC are presented. While the continued slow growth could reflect low level expression of the transporter (at levels below detection in immunoblotting, Figure 2A), a more cautious view might be that the transporter is not strictly essential. This would be more in line with NCR1 in other organisms.

a) A control transfection with the CRISPR guide used for knockout would confirm that the sgRNA used is effective and that target site is accessible.

b) Does continued growth without aTC lead to adaptation and more rapid expansion?

c) Other conditional knockout strategies such as diCre-mediated disruption of the gene (Scientific Reports 7:3881, 2017) could be used to implicate essentiality more conclusively because that approach would not allow continued low-level expression of NCR1 upon rapamycin-induced disruption of the gene.

4) Less important points:

a) Figure 2D, showing a dose-effect for aTC and sensitivity to inhibitor was only done once. I do not consider it appropriate to include data from an experiment performed only once. Do the selected aTC concentrations correspond to measurable differences in PfNCR1 expression, as detected by immunoblotting?

b) Figures 1D-F and 2E-H are critical for linking the mutations in PfNCR1 to action of the three MMV drug leads. Each of these panels shows a < 10-fold change in drug IC50 when the transporter is mutated or knocked down. For each, a single experiment with three replicates is shown with the legends stating that nearly all were only done twice. As variation in parasite growth inhibition experiments is greater between trials (often 2-3 fold in most workers hands) than within a trial, a statistical analysis taking at least 3 independent trials seems necessary. (To my eye, Figures 1D and 1E appear to show only a 2-3 fold difference between WT and mutant.)

c) How many times were the drug selections to identify PfNCR1 performed? What other genes were found in the genome sequencing? Prior studies, e.g. those implicating PfATP4, were more transparent.

d) Figure 1—figure supplement 1B, Southern blot. Each mutant lane shows two bands at 1.3 and 1.1 kB for the retained plasmid and the desired integrant. Shouldn't there also be a higher MW band for the displaced WT locus, as shown in the panel A? Because the mutations shown through DNA sequencing in panel C could derive from the retained plasmid, inclusion of PCR checks for integration and possible residual WT locus in the cloned parasites would help convince careful readers.

Almost no statistical analyses are provided, concerning especially for Figure 1D-F, Figure 2D-H, and Figure 8, where experiments were typically done only once or twice. Each panel shows only the results from a representative experiment, so readers are inclined to believe the better of two trials is shown. Additional trials and statistical analyses are needed to support several conclusions critical to the story.

[Editors' note: further revisions were suggested, as described below.]

Thank you for resubmitting your work entitled "*Plasmodium* Niemann-Pick Type C1-Related Protein is a Druggable Target Required for Parasite Membrane Homeostasis" for further consideration at *eLife*. Your revised article has been favorably evaluated by Anna Akhmanova (Senior Editor), a Reviewing Editor, and three reviewers.

The manuscript has been improved but there are some remaining issues we would recommend addressing before publication, as outlined below:

1) To address reviewer 2 and point 1 of reviewer 3, please ensure that the data are presented across all experiments, and avoid showing representative experiments with errors of technical replicates. Carefully check that no details are missing in the source data files.

2) In regard to point 2 and 3 raised by reviewer 3 you are encouraged to more cautiously interpret your results as suggested.

3) Please take note of the other reviewer comments and address them where possible, or explain why this is not possible.

*Reviewer #1:*

This interesting and high-quality submission is clearly improved from the first submission, as a result of improving the data robustness and experimental descriptions. The authors have provided a robust and suitable response to the reviewers' concerns. I have no further requests for modification and am enthusiastic about the scientific value and broad interest of this report.

*Reviewer #2:*

This is an interesting and high quality study. Improvements have been made to the manuscript.

In some instances there remains a lack of clarity with regards to what data are shown and how many experiments have been performed. The manuscript would also benefit from statistical analyses such as ANOVAs to (for example) compare the sensitivity of different parasite lines to compounds.

Minor Comments:

Information on how many experiments were performed is missing in the legend for Figure 1—figure supplement 1 D and Figure 1—figure supplement 3.

Additional details would be useful in the source data files. What does 'replicates' mean and are they technical replicates, biological replicates or both? Have data from different clones been averaged together?

"One representative experiment with biological triplicates is shown". The term "biological triplicates" is confusing in this context. If replicates were included within an experiment, they should be referred to as 'technical replicates'. 'Biological replicates' implies independent repeats of experiments.

Figure legends should not only state how many times an experiment was performed, but should also specify what the data shown represent. For example Figure 4B-D – it is stated that the experiments were performed twice or three times. However it is not stated whether the data shown in the figures are from single representative experiments or are the averaged data from multiple experiments. The authors should also review their y axes in these figures – they imply that the maximum intracellular GFP was 1%.

Additional data files and statistical comments:

Statistical analyses (e.g. ANOVAs to determine whether IC50 values for compounds vary between mutants and their parent) have still not been performed. Very little statistical information is supplied in the manuscript.

*Reviewer #3:*

The four main questions raised in prior reviews related to localisation of PfNCR1, the protein's essentiality, PfNCR1 role, and statistical analyses of significance in independent experimental trials.

Unfortunately, all these questions remain inadequately addressed.

1) Statistical significance analyses. The authors' rebuttal states that they performed the experiments many times, including in separate labs, but because of protocol differences that they could not be combined and presented together. Another argument provided is that some experiments are only confirmatory of other findings and were only done twice for this reason. A third argument was that some experiments were too "laborious.…[and] just backed up other data". The authors also state "experiments were done at least twice with two different clones, which is essentially biological quadruplicates", which most reviewers will agree is not up to standard.

Some experiments appear to have been done additional times now to reach an n of 3 but most of those figures still appear to show the results from a single experiment with error bars of technical replicates from that trial. These arguments are considered untenable by most journals now, as defined by a 2014 NIH joint workshop with Nature publishing group and Science (see https://www.nih.gov/research-training/rigor-reproducibility/principles-guidelines-reporting-preclinical-research for specifics; see also the PNAS author guidelines which states "Statistical analyses should be done on all available data and not just on data from a "representative experiment." Statistics and error bars should only be shown for independent experiments and not for replicates within a single experiment").

For this reviewer, the most important problem is that key figures still show results from a single experiment with error bars of technical replicates (Figure 1 D-F, Figure 2B-H, Figure 4B-D), preventing readers from evaluating statistical significance between mutants and parental lines; t tests with P values to attest to significant differences are still not provided in the new statistical Tables. For the growth inhibition studies in Figures 1 and 2, can the authors perform a Student's t test comparing mutant vs. wild-type IC50 values from three independent experiments and provide a P value for each reported difference? It is not adequate to state "The error bars (S.D.) for a representative experiment (biological triplicates) are shown and are very small" because these "very small" error bars only inform on technical issues such as pipetting of cells and reagents.

2) PfNCR1 localisation. The split-GFP reporter assay, though creative and challenging to create in cultured parasites, does not unambiguously localize PfNCR1 to the PPM as opposed to the PVM. This is not because of a "very weak signal" as the authors' rebuttal states, but because the split-GFP approach is simply not empowered to address this question. As described in my prior review, a protein at the PVM with C-terminus facing erythrocyte cytosol will produce the same pattern of GFP fluorescence as one at the PPM with C-terminus facing parasite cytosol: both localisations will yield no signal with GFP1 10 targeted to the vacuolar space between these membranes; a "very weak signal" as the authors acknowledge reduces the sensitivity for detecting the other two orientations as well (PVM or PPM proteins with the C-termini facing vacuolar space). The question should either be conclusively addressed or the authors may want to interpret more cautiously.

3) Essentiality of PfNCR1. The authors continue to present PfNCR1 as essential based on failed CRIPSR knockout, dangerous as there are examples where this has yielded incorrect interpretation (e.g. JBC 286:41312 from some of the present authors, later shown to be dispensable PNAS 112:10216 using P. berghei). Rather than adding new data, the authors cite the Zhang et al. genome-wide knockout study as supporting essentiality. Since the authors show expansion of a knockdown line with undetectable PfNCR1 expression and orthologs in other organisms have been proven nonessential, they may again want to interpret more cautiously.

4) Presumed role of PfNCR1. The paper lacks substantive data regarding the proposed role of PfNCR1 in cholesterol or lipid transport. The saponin-sensitivity experiments are interesting, but very similar results are seen with PfATP4 knockdown or block. The author's response "The reviewer's point is well taken" is not matched by new experimentation; the suggestion for complementation studies as used for ScNCR1 and TgNCR1 was dismissed as potentially misleading. The authors rely entirely on computational analysis to assign a role in lipid transport for PfNCR1 and not for PfATP4. If both the role and essentiality were established without the present studies (by computational analysis and the Zhang study), this paper may not meet *eLife* novelty standards.

Minor points:

The anti-HA immuno-EM is good, but IFA as requested in the prior review will be more sensitive for excluding a DV localisation, which remains likely given where NCR1 orthologs localise and the observed changes in DV phenotypes. The interpretation that PfNCR1 changes PPM lipid composition so drastically that endosome and eventually the DV have altered properties as a byproduct is a complicated model that reads as too speculative in the absence of biochemical studies.

Loop-out of the 2nd promoter-less copy, as invoked to account for lack of an expected second band in the Southern blot, is surprising and may be unprecedented in *P. falciparum* as this would require non-homologous end-joining machinery.

---

## [Author Response]

*As noted in the reviews below, the reviewers agree that the findings are of broad interest and suitable novelty for presentation in* eLife*. However, they identified major weaknesses and expect that you will be able to address their various concerns described below by revising the work.*

All of the reviewers have raised serious concerns about statistical significance as it relates to several critical findings. In addition, the interpretation of NCR1 localization and its presumed role and essentiality are insufficiently supported by the data presented.

We thank the editors for considering our manuscript for publication and the reviewers for their careful evaluations.

We agree with the reviewers that it is important to ensure reproducibility. Each assertion in the paper has evidence from multiple lines of experimentation, done multiple times and often reproduced in two different labs. Often an experiment was done many times, but since there were some protocol variations, could not be counted as repetitions even though they support the point being made. Experiments that were done fewer than 3 times were laborious experiments that just backed up other data. Now, most experiments have been done at least three times, and the few that were not have been noted and justified below. The results are very clear and inspire high level of confidence.

The issues of NCR1 localization, role and essentiality are all addressed below.

Separate reviews (please respond to each point):

Reviewer #1:

[…] This is a focused and well executed study. I found the manuscript to be clear and well-written and deserving of publication in eLife. Listed below are several suggestions that could benefit the manuscript.1) I could not find the IC50s of the different compounds. Please add these to the manuscript or to a supplementary table.

EC50s of compounds are similar and in the sub-μM range. Numbers and replicates have been provided in Figure 1—source data 1.

2) Many the experiments reported in this manuscript were done in biological duplicates (N = 2). This is not sufficient to claim statistical significance, at least not if the data were part of a major figure in the paper. A third repeat would be desirable (Figure 1 E-F) and might even help data interpretation regarding Figure 4 B-D. Parasite leakiness in response to saponin and MMV009108 seems lower at 50 and 100 nM of compound versus that observed when treated with MMV028038 and MMV019662. Can this effect be quantified as IC50s to allow for easier comparison between different treatments? Is the lower effect of MMV009108 related to a lower general activity of MMV009108 versus the other compounds?

More replicates have been added for Figures 1, 2 and 4. Tables with concentration-response statistics are presented in Figure 1—source data 1 and 2, Figure 2—source data 1, and Figure 4—source data 1. Parasite sensitivity to saponin permeabilization was similar for the three compounds at 50-100nM. Small variations were observed in different experiments, but the sensitivity was always dramatically different in compound-treated parasites compared to untreated parasites.

3) The GFP signal in Figures 3 and 7 is either too diffuse or is not sufficient in labeling organelles of interest in the microscopy images shown, without the presence of other parasite organelle markers. I would propose to either perform co-staining with known organelle markers to better orient the reader, or move the live-cell and fixed IFA images to the supplement. The immuno-electron microscopy images provided are clearer and by themselves appear satisfactory in showing localization and DV deformation.

Confocal microscopy images for Figure 3D are provided. These make the localization much clearer.

Minor:1) Figure 1C – Could you elaborate on how the schematic was generated based on the limited sequence similarity between hNPC1 and PfNCR1? Was this achieved via visual association?

The model was generated by visual examination of the hNCR1 structure, aided by the alignment of hNCR1 aa 580-794 and aa 1083-1253 with PfNCR1 aa 439-662 and aa 1304-1468, and aided by a partial model of C-terminal residues generated by Robetta. These details have been added to the legend of Figure 1.

2) Figure 2—figure supplement 1A – A schematic of the plasmid, the wild type locus and the subsequent recombinant locus would be more informative when interpreting the gel.

A schematic of the CRISPR/Cas9 modification is provided in Figure 2—figure supplement 1A.

3) Figure 2A – The legend says that the harvested parasites were trophozoite stage yet the text in subsection "PfNCR1 is Important for Asexual Parasite Viability and is Targeted by Antimalarial Compounds" of the Results section says they were highly synchronized young rings. Please clarify.

aTc was removed from young rings and parasites were harvested 20 hours later at the trophozoite stage. This has been clarified in the text.

4) Figure 2C – Is there any obvious explanation as to why the complemented strain proliferates less well than the PfNCR1apt with aTc, especially since the legend mentions that the rescue was done using a stably expressed wt PfNCR1 strain? Also, please report the missing R2 in the legend part C for "+aTc".

There is a 2x difference in growth over 2 weeks. We would not claim a significant growth difference without rigorous competition assays, but there could be a minor effect arising from differences in the sites of transposon integration.

The R^2^ for the rescue experiment +aTc has been added in the figure legend.

5) Figure 2 D-H – A key beside the panels would be helpful.

A key has been provided.

6) The different subpanels of Figure 4 B-D are mislabeled in the figure legend.

This has been corrected.

Reviewer #2:

[…] One concern is that straightforward experiments like FACS-based parasite growth assays (Figure 1 D-F, Figure 2 D,F-H) and measurements of GFP release (Figure 4 C-D) were performed fewer than three times, with data from a single experiment shown. It is not stated how many times the Na^+^ experiment shown in Figure 4F, or the growth assays shown in Figure 1—figure supplement 2 and Figure 4—figure supplement 1C, were performed. Experiments such as these should be performed at least three times and averaged data should be shown or averaged IC50/EC50/rate values provided. The low number of biological replicates means that statistical comparisons cannot be (and were not) performed for these data. Very few of the findings in this study (only the data in Figure 7 C,E) are verified as being statistically significant. The authors state that the PfNCR1 K/D parasites are 'minimally hypersensitive' to KAE609. This statement is not justified based on a subtle difference shown in a single experiment.

In Figure 1, experiments were done at least twice with two different clones, which is essentially biological quadruplicates.

The experiments of Figure 2F-H were only done twice, but they are confirmatory of the western blot data and the 3 compounds and knockdown that all hit the same target are confirmatory of each other.

Figure 4C and D were each done twice, but each recapitulates the results of 4B (done three times) and is concordant with multiple western blots.

Figure 4F has now been repeated a third time.

Regarding KAE609 – our phrasing was not very clear. We do not see hypersensitivity to KAE609 under PfNCR1 knockdown. This experiment has also been repeated (N=3).

Tables with number of repetitions and errors are in Figure 1—source data 1-2, Figure 2—source data 1, and Figure 4—source data 1.

The experiment in Figure 8 is technically very demanding. It was done twice in triplicate and, again, the three compounds gave quite similar profiles.

1) It is not clear how the experiments shown in Figure 2 B and C were performed or how the cumulative parasitemia was calculated. What was the starting parasitemia? Were the cultures diluted and supplemented with uninfected RBCs when required?

This has been clarified in the figure legend. Cultures were seeded at 1% parasitemia, and subjected to daily medium changes, and/or dilutions with uninfected RBCs. Cumulative parasitemias were calculated by multiplying with dilution factors.

2) Figure 7B and 7E – it would be good to see a few representative images for each treatment rather than just one so that readers can understand what was categorised as 'diffuse' and what was categorised as 'puncta'.

Additional images are provided in Figure 7—figure supplement 2.

3) It is not clear what the authors mean when they say that the Na^+^ measurements were calibrated 'using an average from 3 independent calibration curves for SBFI'. Do they mean that three sets of calibrations were performed within each experiment? Or were calibration curves obtained in three initial experiments used to calibrate data obtained in subsequent experiments? Given that the precise cell number and cell age distribution (as well as other factors) will affect the fluorescence ratio obtained in a particular experiment, it would not be appropriate to apply calibration data that were obtained in a different experiment.

Figure 4F has been updated and the figure legend has been clarified. An average of three calibration curves was used. Since we report ratiometric values rather than absolute numbers, differences in cell numbers and cell age distributions do not affect the results. Furthermore, the experiment was done with synchronized trophozoites.

4) Can the authors comment (in the manuscript) on how the resistance levels of the transfectant lines compare to the resistance levels of the compound-selected parasites?

Resistance of transfected lines was slightly lower (3-20 fold) than resistance of selected lines (5-50 fold). Most likely, this is due to amplification of *pfncr1* in the selections, as described in reference 12.

5) Can the authors explain (in the manuscript) the rationale for engineering only one of the two mutations selected for by MMV028038 and MMV019662?

Only one mutation per compound was introduced because the goal of the study was to establish that PfNCR1 is targeted by the compounds, rather than to investigate in detail the role of each mutation. N-terminal mutations are more difficult to engineer using the allelic replacement approach.

6) Can the authors comment (in the manuscript) on whether their inability to knock out pfncr1 was consistent with the results obtained in the mutagenesis study by Zhang et al. (Science)?

Essentiality of Plasmodium *ncr1* is confirmed by Zhang et al. and by Bushell et al. (refs. 17 and 18). This has been added to the text.

7) Is it surprising that the PfNCR1 knock-down parasites grew as well as they did? Have the authors attempted to quantify how much PfNCR1 expression still occurs in the absence of aTc?

The knockdown is good (no HA signal in immunoblot, Figure 2A), but may not be enough for a larger phenotype.

8) Figure 1 title needs correcting: 'Mutations in PfNCR1 confer resistance mutations…'

Has been corrected.

9) Figure 1 legend needs correcting: 'resistant mutations'

Has been corrected.

10) Figure 4 legend needs correcting: 'Treated parasites were washed out…'

Has been corrected.

11) Some of the panel letters referred to in the Figure 4 legend are not used in the correct place.

Has been corrected.

12) What does 'SP' refer to in Figure 3? Would 'PV' be more appropriate?

SP refers to the *sera5* signal peptide. This has been clarified in the figure legend.

13) Figure 6 – can HA on the figure be replaced with PfNCR1-HA?

Repetitions have been done and statistical info has been added where appropriate.

There was a lack of statistical information because many experiments were not performed enough times.Reviewer #3:[…] The authors suggest that PfNCR1 is essential, serves a role in cholesterol export and localizes to the parasite plasma membrane, in contrast to a lysosomal localization for the human ortholog.1) Localization of the putative transporter: The authors show that the GFP-tagged PfNCR1 produces a fluorescence signal around the intracellular parasite (Figure 3A). Immuno-EM using an anti-GFP antibody supports this with gold-dot labeling of either the parasite plasma membrane (PPM) or the PVM; as immuno-EM could not distinguish between the PPM and PVM, they used an interesting split-GFP strategy with expression of soluble GFP1-10 and a short GFP11 tag at the C-terminus of PfNCR1. When GFP1-10 was targeted to the parasite cytosol, they observed a diffuse fluorescence signal in the parasite cytosol (Figure 3D); when GFP1-10 is targeted to the lumen of the PV, no signal was observed. They interpret this as supporting a PPM localization with the C-terminus projecting into the parasite cytoplasm. This approach, though creative and well-executed, raises two concerns.a) The results equally well support a PVM localization with the protein C-terminus projecting into host cytosol. In this alternate model, the signal observed with GFP1-10 in parasite cytosol would represent protein in transit from the parasite ER and exported via vesicular trafficking. Although it is not clear how parasite-derived membrane proteins reach the PVM, export from the PPM on exocytic vesicles and subsequent fusion with the PVM is a favorite model of most workers. Because the GFP11 tag would be intravesicular during this transit and then exposed to host cytosol after insertion, a fluorescence signal with GFP1-10 in the PV lumen would not be expected. Thus, the split-GFP strategy used cannot unambiguously distinguish between PPM and PVM localizations. The authors' statement "our model suggests that the relative orientation of the cytosolic regions in these two distantly related proteins is conserved" applies equally well to both the PPM and PVM models, and so does not help resolve. The authors should consider indirect immunofluorescence after erythrocyte membrane permeabilization with tetanolysin to explore a PVM localization, as has been used by other workers (JBC 278:48413, 2003); while a positive result would conclusively support a PVM localization, a negative result would warrant further experimentation.

We appreciate that interpretation of the split GFP images is difficult due to the very weak signal. Additional live GFP, confocal images have been collected on the split GFP parasites (Figure 3D) and confirm PfNCR1 localization at the PPM with the C-terminus exposed to the parasite cytosol.

b) The fluorescence signals shown in Figure 3A (full-length GFP tag on PfNCR1) and Figure 3D (short GFP11 tag on PfNCR1) appear to be noticeably different from one another, with a rim pattern for the full-length GFP tag and a more diffuse cytoplasmic pattern for the short GFP11 tag. This raises the possibility that a large, full-length GFP tag may cause PfNCR1 mis-trafficking to the PPM/PVM. Did the authors attempt IFA with anti-HA and the PfNCR1-apt parasite, which has an acceptably small epitope tag? This reviewer feels this is necessary, in part because the PPM localization contrasts with the more conventional localization of NCR1 to the lysosome in other organisms; the abnormal DV phenotype in the PfNCR1 knockdown could also suggest that the wild-type protein localizes to the DV (the parasite lysosome-equivalent).

It is unlikely that the GFP-tag causes mis-trafficking as improper function of PfNCR1 is expected to give a severe growth defect (as we have shown with the PfNCR1 knockdown). We have taken additional immune electron micrographs, staining for the C-terminal, single HA epitope present in PfNCR1^apt^ parasites (Figure 3—figure supplement 1). The anti-HA micrographs confirm localization at the parasite surface.

2) Presumed transport activity: The authors name the putative transporter NCR1 based on weak homology to the human NCR1 and orthologs in Saccharomyces and Toxoplasma; I would caution against assigning a name and role based on computational analysis alone. A role in cholesterol transport for ScNCR1 and TgNCR1 were supported by complementation experiments in mammalian cells lacking NPC1. The authors present an increased saponin sensitivity of the parasite upon PfNCR1 knockdown or inhibitor treatment as evidence for cholesterol transport, but they made the same observations in PfATP4 studies in a prior publication. Why didn't they propose a role in cholesterol transport for that transporter? A more cautious view might be that increased saponin sensitivity is a nonspecific parasite response to induced defects at the PPM. Complementation experiments such as performed for ScNCR1 and TgNCR1 appear to be critical for supporting their model. Control knockdowns followed by saponin-sensitivity experiments for other membrane proteins at the PPM may also help to exclude a nonspecific parasite response.

The reviewer’s point is well taken. We assume that the PfNCR1 K/D (or compound treatment) phenotype is due to a defect in function as lipid transporter based on homology. There are multiple elements conserved between hNPC1 and the TgNCR1, and each is found in PfNCR1. We interpret the complementation of ScNCR1 and TgNCR1 to mean that the NCR1 homologs can, when overexpressed, recognize and get enough cholesterol out to prevent toxic accumulation. This does not mean that cholesterol is the natural substrate and for the toxo protein, the data suggest it is not. We would argue that the protein is a Niemann-Pick C1 homolog, hence the Niemann-Pick C1-related protein designation. Proof that PfNCR1 is an actual lipid transporter will await in vitro reconstitution experiments. A discussion of the uncertainties in functional assignment has been added to the Discussion.

3) Essentiality: The authors emphasize that, in contrast to presumed orthologs in other organisms, PfNCR1 is essential. This seems to be based on negative results with attempted CRISPR knockout and on a delayed slow growth phenotype with conditional knockdown using the TetR-DOZI/aptamer system. Remarkably, the delayed slow growth phenotype appears to take ~ 10 days (about 5 asexual cycles) to be manifest clearly as no statistics on differences in growth rate for plus and minus aTC are presented. While the continued slow growth could reflect low level expression of the transporter (at levels below detection in immunoblotting, Figure 2A), a more cautious view might be that the transporter is not strictly essential. This would be more in line with NCR1 in other organisms.a) A control transfection with the CRISPR guide used for knockout would confirm that the sgRNA used is effective and that target site is accessible.b) Does continued growth without aTC lead to adaptation and more rapid expansion?c) Other conditional knockout strategies such as diCre-mediated disruption of the gene (Scientific Reports 7:3881, 2017) could be used to implicate essentiality more conclusively because that approach would not allow continued low-level expression of NCR1 upon rapamycin-induced disruption of the gene.

We believe it is likely that PfNCR1 performs an essential function based on the observation that the K/D slows parasite replication and that the three compounds which kill parasites are on target. Whole genome disruption analyses are consistent with this, and now have been cited in the text.

Continued growth without aTc results in loss of aptamers eliminating the knockdown effect.

4) Less important points:a) Figure 2D, showing a dose-effect for aTC and sensitivity to inhibitor was only done once. I do not consider it appropriate to include data from an experiment performed only once. Do the selected aTC concentrations correspond to measurable differences in PfNCR1 expression, as detected by immunoblotting?

It had been done multiple times in different forms, but the exact experiment has now been done three times. Yes, aTc concentrations correspond to changes in PfNCR1 expression.

b) Figures 1D-F and 2E-H are critical for linking the mutations in PfNCR1 to action of the three MMV drug leads. Each of these panels shows a < 10-fold change in drug IC50 when the transporter is mutated or knocked down. For each, a single experiment with three replicates is shown with the legends stating that nearly all were only done twice. As variation in parasite growth inhibition experiments is greater between trials (often 2-3 fold in most workers hands) than within a trial, a statistical analysis taking at least 3 independent trials seems necessary. (To my eye, Figures 1D and 1E appear to show only a 2-3 fold difference between WT and mutant.)

Number of replicates and changes in EC_50_s for experiments in Figures 1D-F and 2E-H are reported as supplemental tables.

c) How many times were the drug selections to identify PfNCR1 performed? What other genes were found in the genome sequencing? Prior studies, e.g. those implicating PfATP4, were more transparent.

All drug selections were performed at least in triplicate. Selection protocols and complete sequencing of resistant clones has been reported in reference 12. In addition to the resistance-conferring SNPs, amplification of the *pfncr1* locus was observed.

d) Figure 1—figure supplement 1B, Southern blot. Each mutant lane shows two bands at 1.3 and 1.1 kB for the retained plasmid and the desired integrant. Shouldn't there also be a higher MW band for the displaced WT locus, as shown in the panel A? Because the mutations shown through DNA sequencing in panel C could derive from the retained plasmid, inclusion of PCR checks for integration and possible residual WT locus in the cloned parasites would help convince careful readers.

Diagnostic PCRs in Figure 1—figure supplement 1C and Figure 1—figure supplements 2A, B illustrate that the sequencing data derive from integrated mutations. The reviewer is correct that a second, higher MW band (corresponding to the 2^nd^, promoter-less copy) should be present in the Southern blot of the integrants used in Figure 1. The 2^nd^ copy looped out in some of our clones. In these clones, we have shown that the mutation is present at the endogenous expression site. We also obtained clones in which the promoter-less copy is present and show in Figure 1—figure supplement 1D that clones with or without the 2^nd^ copy are resistant. Importantly, the wild-type copy is disrupted in all allelic replacement clones as illustrated in Figure 1—figure supplement 1B, C.

Almost no statistical analyses are provided, concerning especially for Figure 1D-F, Figure 2D-H, and Figure 8, where experiments were typically done only once or twice. Each panel shows only the results from a representative experiment, so readers are inclined to believe the better of two trials is shown. Additional trials and statistical analyses are needed to support several conclusions critical to the story.

Experiments have been repeated and statistical tables are provided for Figures 1, 2, and 4.

[Editors' note: further revisions were suggested, as described below.]

Reviewer #2:

[…] Minor Comments:Information on how many experiments were performed is missing in the legend for Figure 1—figure supplement 1 D and Figure 1—figure supplement 3.

The number of replicates in Figure 1—figure supplement 1D and Figure 1—figure supplement 3 was 2. This information has been added to the figure legends.

Additional details would be useful in the source data files. What does 'replicates' mean and are they technical replicates, biological replicates or both? Have data from different clones been averaged together?

T-tests have been added for data in the source data files. Yes, data from different clones have been averaged together in the source data files.

"One representative experiment with biological triplicates is shown". The term "biological triplicates" is confusing in this context. If replicates were included within an experiment, they should be referred to as 'technical replicates'. 'Biological replicates' implies independent repeats of experiments.

This sentence refers to Figures 2B and 2C. We corrected the statement in the figure legend to now says “One representative experiment with technical triplicates is shown.”

Figure legends should not only state how many times an experiment was performed, but should also specify what the data shown represent. For example Figure 4B-D – it is stated that the experiments were performed twice or three times. However it is not stated whether the data shown in the Figures are from single representative experiments or are the averaged data from multiple experiments. The authors should also review their y axes in these Figures – they imply that the maximum intracellular GFP was 1%.

Figure 4 panels are from a single representative experiment with technical duplicates. This is now stated in the text. The y axes have been changed.

Additional data files and statistical comments:Statistical analyses (e.g. ANOVAs to determine whether IC50 values for compounds vary between mutants and their parent) have still not been performed. Very little statistical information is supplied in the manuscript.

Statistical analyses are now included source data files.

Reviewer #3:

The four main questions raised in prior reviews related to localisation of PfNCR1, the protein's essentiality, PfNCR1 role, and statistical analyses of significance in independent experimental trials.Unfortunately, all these questions remain inadequately addressed.1) Statistical significance analyses. The authors' rebuttal states that they performed the experiments many times, including in separate labs, but because of protocol differences that they could not be combined and presented together. Another argument provided is that some experiments are only confirmatory of other findings and were only done twice for this reason. A third argument was that some experiments were too "laborious.…[and] just backed up other data". The authors also state "experiments were done at least twice with two different clones, which is essentially biological quadruplicates", which most reviewers will agree is not up to standard.Some experiments appear to have been done additional times now to reach an n of 3 but most of those figures still appear to show the results from a single experiment with error bars of technical replicates from that trial. These arguments are considered untenable by most journals now, as defined by a 2014 NIH joint workshop with Nature publishing group and Science (see https://www.nih.gov/research-training/rigor-reproducibility/principles-guidelines-reporting-preclinical-research for specifics; see also the PNAS author guidelines which states "Statistical analyses should be done on all available data and not just on data from a "representative experiment." Statistics and error bars should only be shown for independent experiments and not for replicates within a single experiment").

We take issue with this criticism. Two replicates with two clones (in technical triplicate) and a wealth of other data is arguably better than triplicates consisting of a single clone. T-tests have been done on the data in all experiments in the source data.

For this reviewer, the most important problem is that key figures still show results from a single experiment with error bars of technical replicates (Figure 1 D-F, Figure 2B-H, Figure 4B-D), preventing readers from evaluating statistical significance between mutants and parental lines; t tests with P values to attest to significant differences are still not provided in the new statistical Tables. For the growth inhibition studies in Figures 1 and 2, can the authors perform a Student's t test comparing mutant vs. wild-type IC50 values from three independent experiments and provide a P value for each reported difference? It is not adequate to state "The error bars (S.D.) for a representative experiment (biological triplicates) are shown and are very small" because these "very small" error bars only inform on technical issues such as pipetting of cells and reagents.

T-tests have been done on the data in all experiments in the source data.

2) PfNCR1 localisation. The split-GFP reporter assay, though creative and challenging to create in cultured parasites, does not unambiguously localize PfNCR1 to the PPM as opposed to the PVM. This is not because of a "very weak signal" as the authors' rebuttal states, but because the split-GFP approach is simply not empowered to address this question. As described in my prior review, a protein at the PVM with C-terminus facing erythrocyte cytosol will produce the same pattern of GFP fluorescence as one at the PPM with C-terminus facing parasite cytosol: both localisations will yield no signal with GFP1 10 targeted to the vacuolar space between these membranes; a "very weak signal" as the authors acknowledge reduces the sensitivity for detecting the other two orientations as well (PVM or PPM proteins with the C-termini facing vacuolar space). The question should either be conclusively addressed or the authors may want to interpret more cautiously.

The weak signal made it look more diffuse. The rim pattern of the new confocal images suggests that the GFP11 on PfNCR1 is not contained in vesicles in the cytoplasm. We have added the following sentence in the text: “We cannot rule out the possibility that the cytoplasmic GFP 1-10 signal is due to vesicles at the PPM in transit to the PVM.”

3) Essentiality of PfNCR1. The authors continue to present PfNCR1 as essential based on failed CRIPSR knockout, dangerous as there are examples where this has yielded incorrect interpretation (e.g. JBC 286:41312 from some of the present authors, later shown to be dispensable PNAS 112:10216 using P. berghei). Rather than adding new data, the authors cite the Zhang et al. genome-wide knockout study as supporting essentiality. Since the authors show expansion of a knockdown line with undetectable PfNCR1 expression and orthologs in other organisms have been proven nonessential, they may again want to interpret more cautiously.

Our logic is: 1) the drugs kill parasites; 2) K/D phenocopies drug treatment; 3) K/D is hypersensitive to drugs. We believe that these observations strongly suggest that PfNCR1 is essential. In deference to the reviewer we softened the wording in the Discussion. Also, it should be noted that essentialities of several genes are known to vary among different species (as seen in the instance cited by the reviewer; even in this instance, there was a clear growth defect in P. berghei for the ATP synthase subunit knockout, something that could not be achieved for *P. falciparum*, which we have now attempted numerous times including with the CRISPR technology).

4) Presumed role of PfNCR1. The paper lacks substantive data regarding the proposed role of PfNCR1 in cholesterol or lipid transport. The saponin-sensitivity experiments are interesting, but very similar results are seen with PfATP4 knockdown or block. The author's response "The reviewer's point is well taken" is not matched by new experimentation; the suggestion for complementation studies as used for ScNCR1 and TgNCR1 was dismissed as potentially misleading. The authors rely entirely on computational analysis to assign a role in lipid transport for PfNCR1 and not for PfATP4. If both the role and essentiality were established without the present studies (by computational analysis and the Zhang study), this paper may not meet eLife novelty standards.

We maintain that complementation studies will not answer the question. We have stated in the Discussion that “Proof of a direct role of PfNCR1 as a lipid transporter awaits functional analysis.”

Minor points:The anti-HA immuno-EM is good, but IFA as requested in the prior review will be more sensitive for excluding a DV localisation, which remains likely given where NCR1 orthologs localise and the observed changes in DV phenotypes. The interpretation that PfNCR1 changes PPM lipid composition so drastically that endosome and eventually the DV have altered properties as a byproduct is a complicated model that reads as too speculative in the absence of biochemical studies.

We believe cryoEM is better suited to address the localization because of better access to the DV, as long as there is good signal. Antibody access to the DV can be an issue with IFA.

Loop-out of the 2nd promoter-less copy, as invoked to account for lack of an expected second band in the Southern blot, is surprising and may be unprecedented in P. falciparum as this would require non-homologous end-joining machinery.

Integration of multiple concatamerized plasmids is common in Plasmodium. We also observe multiple plasmid copies as is evident on the Southern blot. We believe that the expected promoterless second copy in the mutation integrant parasites used in Figure 1 has looped out via homologous crossover after the BspHI site in the integrated concatamerized plasmid. This mechanism does not require non-homologous end-joining. We have edited the figure legend to better explain this possible mechanism for the looping out.